# UBTD1 regulates ceramide balance and endolysosomal positioning to coordinate EGFR signaling

Stéphanie Torrino[1,2]*, Victor Tiroille[2†], Bastien Dolfi[3†], Maeva Dufies[4], Charlotte Hinault[2,5], Laurent Bonesso[5], Sonia Dagnino[6], Jennifer Uhler[7], Marie Irondelle[8], Anne-sophie Gay[1], Lucile Fleuriot[1], Delphine Debayle[1], Sandra Lacas-Gervais[9], Mireille Cormont[10], Thomas Bertero[1], Frederic Bost[2]*, Jerome Gilleron[10‡], Stephan Clavel[2‡]*

[1]Université Côte d'Azur, CNRS, IPMC, Valbonne, France; [2]Université Côte d'Azur, Inserm, C3M, Team Targeting prostate cancer cell metabolism, Nice, France; [3]Université Côte d'Azur, Inserm, C3M, Team Metabolism and cancer, Nice, France; [4]Biomedical Department, Centre Scientifique de Monaco, Monaco, Monaco; [5]Biochemistry Laboratory, University Hospital, Nice, France; [6]MRC Centre for Environment and Health, Department of Epidemiology and Biostatistics, School of Public Health, Imperial CollegeLondon, London, United Kingdom; [7]Department of Medical Biochemistry and Cell Biology, University of Gothenburg, Gothenburg, Sweden; [8]Inserm U1065, Université Côte d'Azur, Nice, France; [9]CCMA, UFR Sciences, Université Côte d'Azur, Nice, France; [10]Université Côte d'Azur, Inserm, C3M, Team Cellular and Molecular Pathophysiology of Obesity and Diabetes, Nice, France

*For correspondence:
stephanie.torrino@unice.fr (ST);
Frederic.BOST@univ-cotedazur.fr
(FB);
Stephan.CLAVEL@univ-cotedazur.
fr (SC)

†These authors contributed
equally to this work
‡These authors also contributed
equally to this work

Competing interests: The
authors declare that no
competing interests exist.

Reviewing editor: Roger J
Davis, University of
Massachusetts Medical School,
United States

**Abstract** To adapt in an ever-changing environment, cells must integrate physical and chemical signals and translate them into biological meaningful information through complex signaling pathways. By combining lipidomic and proteomic approaches with functional analysis, we have shown that ubiquitin domain-containing protein 1 (UBTD1) plays a crucial role in both the epidermal growth factor receptor (EGFR) self-phosphorylation and its lysosomal degradation. On the one hand, by modulating the cellular level of ceramides through N-acylsphingosine amidohydrolase 1 (ASAH1) ubiquitination, UBTD1 controls the ligand-independent phosphorylation of EGFR. On the other hand, UBTD1, via the ubiquitination of Sequestosome 1 (SQSTM1/p62) by RNF26 and endolysosome positioning, participates in the lysosomal degradation of EGFR. The coordination of these two ubiquitin-dependent processes contributes to the control of the duration of the EGFR signal. Moreover, we showed that UBTD1 depletion exacerbates EGFR signaling and induces cell proliferation emphasizing a hitherto unknown function of UBTD1 in EGFR-driven human cell proliferation.

## Introduction

All living organisms perceive variations in their environment and translate them into intracellular signals via signaling pathways. In multicellular organisms, disturbances in this signal transduction mechanism induce inappropriate cell behavior and are associated with a plethora of diseases including cancer.

Cellular signaling can be viewed as a finely tuned 'space-time continuum' (*Scott and Pawson, 2009*). Receptors activated by their ligands at the plasma membrane are endocytosed, then moved

along endocytic compartments to be routed to the lysosomes for final degradation or recycled back to the cell surface. Thus, the signal delivered to the cell is the sum of the signals emitted by the activated receptor at the plasma membrane and during its intracellular trafficking (*Bakker et al., 2017*). The responsiveness of these processes requires fast and accurate control in space and time, which is mainly ensured by post-translational modification (PTM) of proteins. Broadly, several aspects of cell signaling and receptor trafficking are regulated by proteolytic or non-proteolytic ubiquitination (*Bakker et al., 2017*; *Rape, 2018*; *Haglund and Dikic, 2005*; *Tokunaga et al., 2009*; *Eden et al., 2012*).

Protein ubiquitination is a PTM that results in the covalent attachment of one or more ubiquitin to lysine residues of the substrate (*Yau and Rape, 2016*; *Komander and Rape, 2012*; *Kerscher et al., 2006*). Ubiquitin conjugation occurs in a sequential three-step enzymatic process involving E1 (ubiquitin activation), E2 (ubiquitin conjugation), and E3 (ubiquitin ligation). In this hierarchical framework, there are only two E1s, over 30 E2s and hundreds of E3s in human, illustrating the large spectrum of E2s that contrasts with the high specificity of E3s in the recognition of substrates (*Schulman and Harper, 2009*; *Ye and Rape, 2009*). While the interaction of E3s with their substrates confers specificity to the system, E2s are more versatile and are commonly considered as 'ubiquitin carriers' with an auxiliary rather than control role. However, the modulation of the functionality of E2/E3 complexes by scaffold proteins has been poorly investigated and few proteins acting on these complexes have been characterized (*Ye and Rape, 2009*; *Good et al., 2011*).

Ubiquitin domain-containing protein 1 (UBTD1) is an evolutionarily-conserved protein which interacts, both in vitro and in vivo, with some E2 and E3 enzymes of the ubiquitin-proteasome system (UPS) (*Uhler et al., 2014*). Recently, we have shown that UBTD1 controls the degradation of the transcriptional regulator yes-associated protein (YAP) by modulating its ubiquitination (*Torrino et al., 2019*). Mechanistically, UBTD1 is a component of the ubiquitination complex that allows the E3 ligase beta-transducin repeats-containing proteins (β-TRCP) to interact with YAP. Coherent with this E2/E3 regulatory function, UBTD1 has been reported to stabilize p53 (tumor protein P53) through ubiquitination and further degradation of mouse double minute 2 (MDM2), the E3 enzyme that degrades p53 (*Zhang et al., 2015*). Importantly, a decreased expression of UBTD1 is associated with increased cancer aggressiveness and decreased overall patient survival in colorectal, liver, prostate, and lung cancer (*Uhler et al., 2014*; *Torrino et al., 2019*; *Yang et al., 2020*). Based on these evidences, we have previously suggested that UBTD1 may be an E2/E3 scaffolding protein acting as a tumor suppressor (*Torrino et al., 2019*).

To provide novel insights into UBTD1 functions, we here investigated the effect of its knockdown in a cell model by combining lipidomic, proteomic, and signaling screening. Through this integrated approach, we have uncovered an important role of UBTD1 in epidermal growth factor receptor (EGFR) degradation and signaling. Indeed, we showed that UBTD1 participates in the processing of EGFR-positive vesicles and controls late endosome/lysosome positioning through p62/SQSTM1 (Sequestosome 1) ubiquitination by RNF26. UBTD1 also regulates the ubiquitination of lysosomal ceramidase N-Acylsphingosine Amidohydrolase 1 (ASAH1) and modifies membrane lipid composition to limit EGFR auto-phosphorylation. UBTD1 coordinates in time and space the EGFR signaling pathway to avoid persistent and inappropriate signaling leading to uncontrolled cell proliferation.

## Results

### UBTD1 depletion induces EGFR self-phosphorylation by modifying membrane lipid composition through ASAH1 ubiquitination

In mammals, UBTD1 is expressed in many organs and cell types, however its molecular function is still largely unknown, and its cellular role remains elusive. To provide insights into its cellular functions, we first performed a phospho-kinase array allowing us to observe that UBTD1 depletion increased (>1.5 fold) phosphorylation of JNK1/2/3, GSK3 α/β (S9/21) and EGFR (Y1086) (*Figure 1A*, *Figure 1—figure supplement 1A*). EGFR phosphorylation presented the highest difference compared to control cells (>2 fold). Next, we confirmed by western-blot an increase in EGFR phosphorylation (Y1068 and Y1086) at steady state, and we also noticed an increase in the total amount of EGFR in UBTD1-depleted cells (*Figure 1B*, *Figure 1—figure supplement 1B*). Coherently with an increased EGFR signaling, UBTD1 depletion sharply increases cell proliferation (*Figure 1C*). These

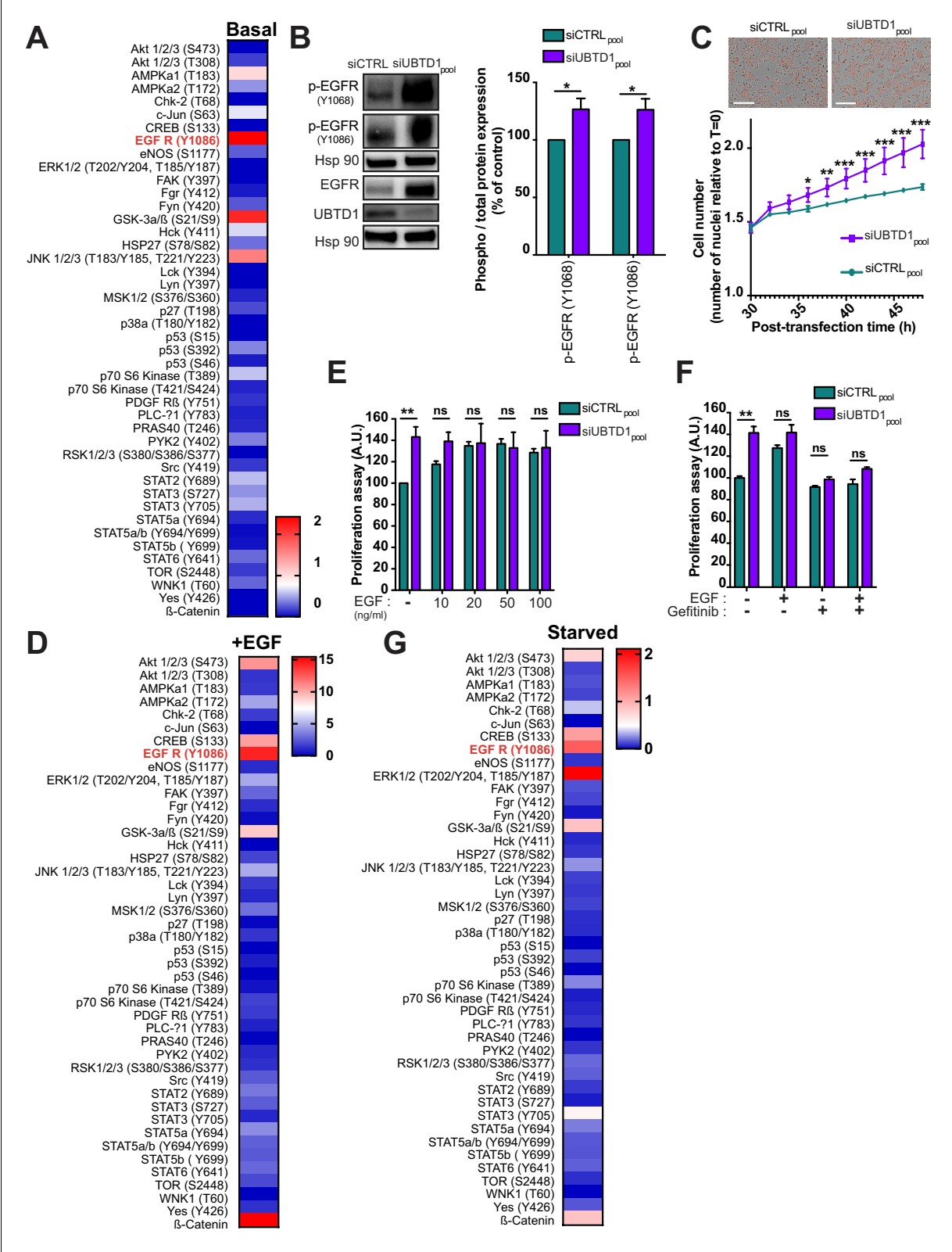

**Figure 1.** UBTD1 depletion exacerbates EGFR signaling to induce cell proliferation. (A–G) DU145 cells were transfected for 48 hr with the indicated siRNA (siCTRLpool or siUBTD1pool). (A, D, G) Heatmap of protein phosphorylation relative quantification from multiple signaling pathways in complete media (A), under EGF stimulation (D) at 50 ng/ml for 10 min or under serum-starved medium (G). Data quantification is carried out from dot blot (*Figure 1—figure supplement 1A*, C, H) and normalized to the siCTRL condition. (B) Immunoblot and quantification of p-EGFR (Y1068 or Y1086).

*Figure 1 continued on next page*

*Figure 1 continued*

p-EGFR levels were quantified by calculating the ratio between p-EGFR and EGFR, both normalized to loading control signal. Immunoblot of UBTD1 shows the level of siRNA depletion. (C) Representative images and cell growth curves measured by videomicroscopy (Incucyte, Essen Bioscience). T = 0 corresponds to transfection time and the time window (30–50 hr) is presented. (E, F) Proliferation assay in the presence of different concentrations of EGF (E) and in the presence of EGF inhibitor (Gefitinib) (F). n ≥ 3 independent experiments. ns = non-significant, *p<0.05; **p<0.01, ***p<0.001; (C) Bonferroni's multiple comparison test; (E–F) two-way ANOVA and Bonferroni's multiple comparisons test; data are mean ± s.e.m.

The online version of this article includes the following source data and figure supplement(s) for figure 1:

**Source data 1.** Uncropped western blot for *Figure 1*.
**Source data 2.** Row data for *Figure 1* and for *Figure 1—figure supplement 1*.
**Figure supplement 1.** UBTD1 depletion exacerbates EGFR signaling.
**Figure supplement 1—source data 1.** Uncropped western blot for *Figure 1—figure supplement 1*.

observations led us to repeat the same experiment under EGF stimulation (*Figure 1D*, *Figure 1—figure supplement 1C*). As compared to EGF-treated-control cells, UBTD1 depletion drastically increased (5 to 20-fold) the phosphorylation of PKB/Akt, β-catenin, GSK3 α/β, CREB, ERK 1/2, and EGFR. In UBTD1 depleted cells, the phosphorylation status of EGFR is the most dramatically changed (15-20-fold), suggesting a close link between UBTD1 and EGFR. We then carefully evaluated, at steady state, the activation of the signaling pathways downstream of the EGFR (*Figure 1—figure supplement 1D–G*; *Avraham and Yarden, 2011*; *Lemmon and Schlessinger, 2010*; *Wells, 1999*; *Grant et al., 2002*). As expected, UBTD1 depletion increased STAT3 phosphorylation, nuclear translocation and STAT3 target gene expression: Bcl-2, HIF-2, and MMP-2,–9. Likewise, the phosphorylation of ERK (T202/Y204) and Akt (S473/T308) are increased in UBTD1-depleted cells. EGF promotes proliferation of control cells in a dose-dependent manner but does not further increase proliferation induced by UBTD1 depletion (*Figure 1E*), while the addition of Gefitinib, an EGFR inhibitor, totally abolishes the pro-proliferative effect of UBTD1 knockdown, indicating that this effect is EGFR-dependent but EGF-independent (*Figure 1F*). These converging results indicate that UBTD1 depletion increases EGFR phosphorylation and amplifies signaling cascades downstream of EGFR leading to increased proliferation.

We next wanted to elucidate the underlying mechanism by which UBTD1 controls EGFR phosphorylation. We performed a phospho-kinase array in serum free medium to determine whether some molecules present in the medium could be involved in EGFR activation (*Figure 1G*, *Figure 1—figure supplement 1H*; *Sigismund et al., 2018*). Surprisingly, although no growth factors were present in the medium, the levels of EGFR phosphorylation and its downstream targets (Akt, ERK, CREB, STATs) were still higher in UBTD1-depleted cells than in control cells. Therefore, we then postulated that UBTD1-depleted cells could secrete factors that elicit EGFR phosphorylation. Besides EGF, several other ligands can bind to and activate EGFR, including TGF-α, heparin-binding EGF-like growth factor (HB–EGF), amphiregulin, betacellulin, epigen, and epiregulin (*Grant et al., 2002*). Thus, we tested whether the knockdown of UBTD1 increases the mRNA expression and/or the secretion of these EGFR ligands. In UBTD1-depleted cells, neither the expression (*Figure 1—figure supplement 1I,K*) nor the secretion (*Figure 1—figure supplement 1J,L*) of the main EGFR ligands was increased. This data led us to consider that UBTD1 depletion induces ligand-independent EGFR phosphorylation.

Membrane lipid composition surrounding tyrosine kinase receptors can modulate their activation (*Coskun et al., 2011*; *Ferguson et al., 2003*; *Meuillet et al., 2000*; *Lambert et al., 2006*; *Todeschini et al., 2008*). Thus, we isolated cell membranes and performed a lipidomic analysis. UBTD1 depletion induces major changes in membrane lipid composition and, notably, altered the ceramide subclass (*Figure 2A–B*, *Figure 2—figure supplement 1A*). Next, by MALDI-TOF/TOF mass spectrometry, which detects the non-lipidic moieties and the identity of the ceramide-backboned lipids, we identified, in UBTD1-depleted cells, a drop in GM2 and GM3 d18:1/16:0 gangliosides (*Figure 2C–D*). The GM3 ganglioside inhibits spontaneous EGFR autophosphorylation (*Coskun et al., 2011*; *Meuillet et al., 2000*; *Bremer et al., 1986*). To test whether EGFR phosphorylation induced by UBTD1 depletion was caused by a decrease in GM3 content, we added GM3 to the culture medium. In UBTD1-depleted cells, addition of GM3 restores, in a dose-dependent manner, the level of EGFR phosphorylation observed in control cells and completely abolishes cell proliferation induced by UBTD1 knockdown (*Figure 2E,F*), suggesting that the increase in EGFR

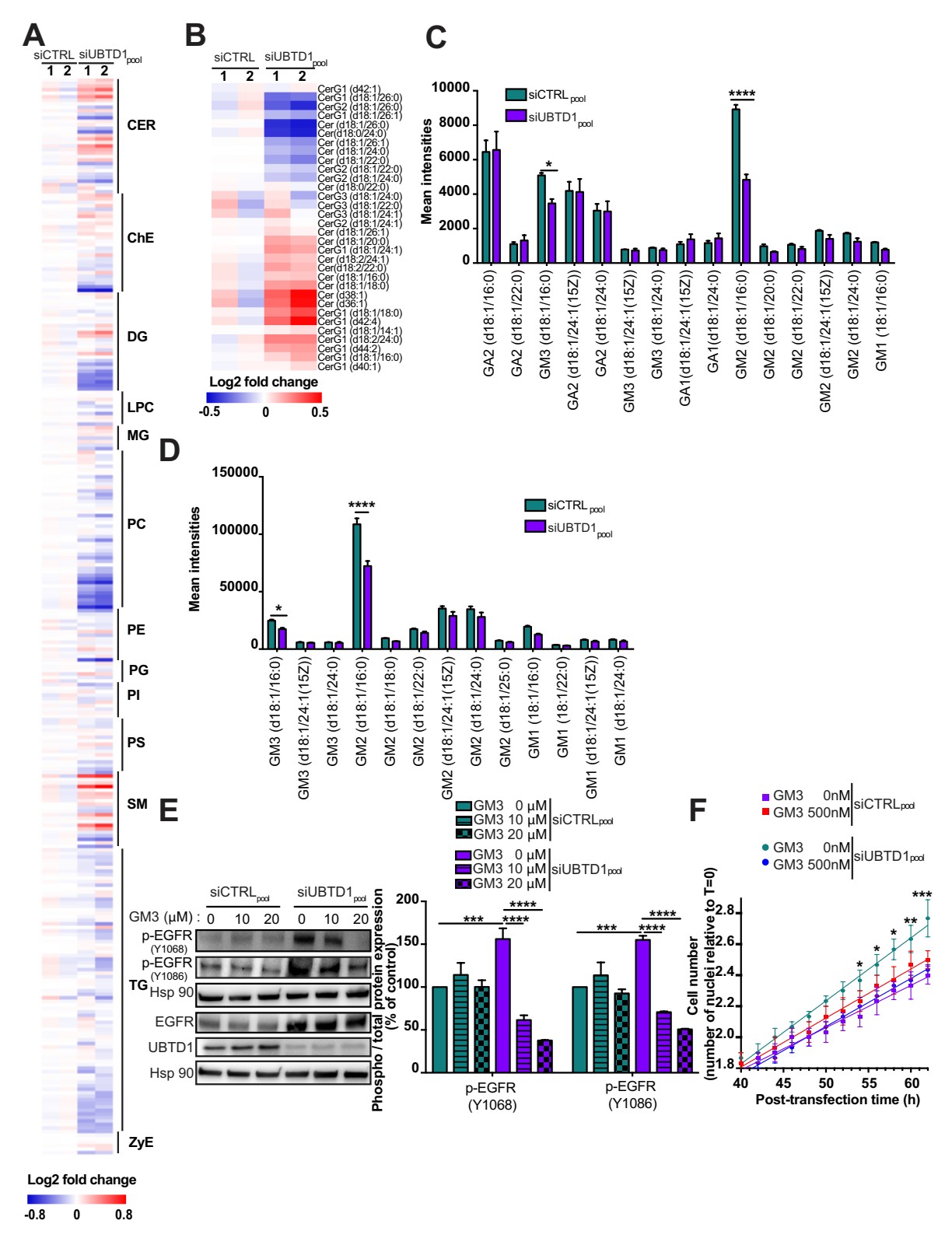

**Figure 2.** UBTD1 depletion induces EGFR self-phosphorylation by modifying membrane lipid composition. (**A–F**) DU145 cells were transfected for 48 hr with the indicated siRNA (siCTRLpool or siUBTD1pool). (**A,B**) Heatmap of all (**A**) lipids and (**B**) ceramide levels. The normalized expression of each lipid is shown in a scale range from blue to red. CER: ceramides; ChE: cholesterol ester; DG: diacylglycerols; LPC: lysophosphatidylcholines; MG: monoacylglycerols; PC: phosphatidylcholines; PE: phosphatidylethanolamines; PG: phosphatidylglycerol; PI: phosphatidylinositol; PS:

*Figure 2 continued on next page*

*Figure 2 continued*

phosphatidylserines; SM: sphingomyelins; TG: triglycerides; ZyE: zymosterols. (**C,D**) Mean intensities of positive-ion (**C**) and negative-ion (**D**) MS spectra from reflectron MALDI-TOF analyses. (**E**) Immunoblot and quantification of p-EGFR (Y1068 or Y1086) in the presence of different concentrations of GM3. p-EGFR levels were quantified by calculating the ratio between p-EGFR and EGFR, both normalized to loading control signal. Immunoblot of UBTD1 shows the level of siRNA depletion. (**F**) Representative cell growth curves measured by videomicroscopy (Incucyte, Essen Bioscience) in the presence of GM3. T = 0 corresponds to transfection time and the time window (40–60 hr) is presented. n ≥ 3 independent experiments. *p<0.05 **p<0.01, ***p<0.001, ****p<0.0001; (**C–F**) two-way ANOVA and Bonferroni's multiple comparisons test; data are mean ± s.e.m.

The online version of this article includes the following source data and figure supplement(s) for figure 2:

**Source data 1.** Uncropped western blot for *Figure 2*.
**Source data 2.** Row data for *Figure 2* and for *Figure 2—figure supplement 1*.
**Figure supplement 1.** Protein clustering and functional interaction analysis of the UBTD1 interactome.
**Figure supplement 1—source data 1.** Uncropped western blot for *Figure 2—figure supplement 1*.

phosphorylation and cell proliferation was due to a drop in GM3. To gain insight on how UBTD1 may alter ganglioside level, we performed an immunoprecipitation experiment using endogenous UBTD1 as a bait. Next, by mass spectrometry we identify UBTD1 partners and associated proteins. Using a mild stringency buffer (NP40) to preserve weak protein interactions, we identified a set of 463 proteins (*Figure 2—figure supplement 1*) distributed in different cell compartments including plasma membrane, endolysosome, and endoplasmic reticulum (ER) (*Figure 2—figure supplement 1*). Then, we generated a UBTD1 interactome and performed an unsupervised cluster analysis to define 14 protein clusters (*Figure 2—figure supplement 1B,C*). By assigning to these clusters their top biological functions, we identify a functional sub-network composed of seven proteins and associated with the ceramide metabolic pathway (*Figure 2—figure supplement 1D*). Within this cluster, we found two proteins directly involved either in the synthesis (ceramide synthase 2, CerS2) or in the degradation (lysosomal ceramidase, ASAH1) of ceramides. Although the protein level of CerS2 was unaffected by UBTD1 invalidation, the level of ASAH1 protein was significantly increased (*Figure 3A*). ASAH1 degradation was found to be controlled by the proteasome in a prostate cancer cell line (*Mizutani et al., 2015*). To test whether UBTD1 regulates the degradation of ASAH1, we blocked protein synthesis with cycloheximide (*Figure 3B*). In the UBTD1-depleted cells, we observed a substantial increase in ASAH1 protein lifetime compared to control cells. Next, we performed a ubiquitination assay (*Figure 3C*). UBTD1 knockdown drastically reduced ASAH1 ubiquitination, supporting that UBTD1 participates in ASAH1 degradation. Finally, to demonstrate that the increase of EGFR phosphorylation induced by UBTD1 depletion occurs through a defect of ASAH1 degradation, we knock down ASAH1 in UBTD1-depleted cells (*Figure 3D*, *Figure 2—figure supplement 1E*). The depletion of ASAH1 does not modify the phosphorylation of EGFR or its signaling as reflected by STAT3 phosphorylation. However, in UBTD1-depleted cells, the knockdown of ASAH1 severely decreases the phosphorylation of EGFR and STAT3 induced by UBTD1 depletion. Collectively, we here provided compelling evidence showing that UBTD1 depletion induces EGFR phosphorylation by decreasing GM3 level through impairment of ASAH1 ubiquitin-dependent degradation.

## UBTD1 is associated with EGFR and delays its lysosomal degradation

Although the change in the amount of ganglioside GM3 convincingly explains the effect of UBTD1 depletion on EGFR phosphorylation, the GM3 supplementation does not rescue the level of total EGFR, suggesting an additional role of UBTD1 on EGFR turnover. Thus, we evaluate the effect of UBTD1 depletion on EGFR turnover by blocking protein synthesis with cycloheximide (*Figure 4A*, *Figure 4—figure supplement 1A*). When treated with EGF, the level of EGFR decreases over time. Conversely, in UBTD1-depleted cells, the amount of EGFR remains constant for, at least, 3 hr after the addition of EGF, showing that UBTD1 acts post-transcriptionally on EGFR.

To decipher the underlying mechanism, we took advantage of the cluster functional analysis generated from the UBTD1 interactome (*Figure 2—figure supplement 1B,C*). Interestingly, we identified an extensive protein cluster functionally related to vesicle mediated transport and endocytosis, which includes EGFR (*Figure 2—figure supplement 1F*). This data provides an important clue to understand how UBTD1 can affect EGFR turnover. Indeed, the internalization of EGFR relies on its ubiquitination and we previously demonstrated that UBTD1 interacts with E2/E3 of the UPS (*Bakker et al., 2017*; *Uhler et al., 2014*; *Caldieri et al., 2018*). Therefore, we next hypothesized

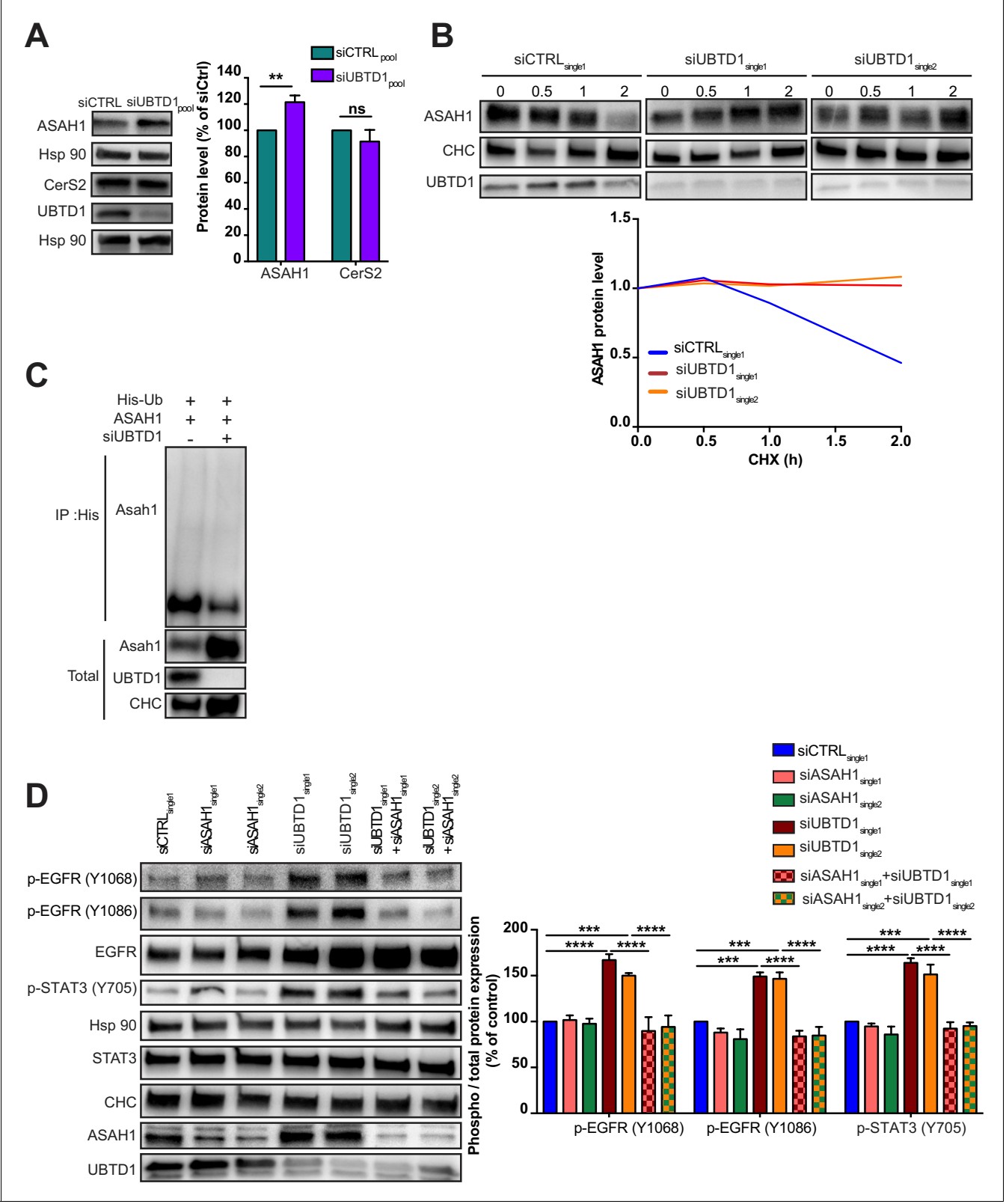

**Figure 3.** UBTD1 controls ASAH1 ubiquitination to promote EGFR self-phosphorylation. (A–D) DU145 cells were transfected for 48 hr with the indicated siRNA (siCTRLpool or siUBTD1pool or siUBTD1single1 or single2 or siASAH1 single1 or single2). (A) Immunoblot and quantification of ceramide synthase 2 (CerS2) and the lysosomal ceramidase (ASAH1). (B) Immunoblots (up) and quantification (down) of ASAH1 levels in cells treated with cycloheximide (CHX) at different time points. Immunoblot of UBTD1 shows the level of siRNA depletion. CHC was used as a loading control. (C)

*Figure 3 continued on next page*

*Figure 3 continued*

Immunoblots show ASAH1 ubiquitylation in HEK cells in different experimental conditions. Cells were transfected, as indicated, with expression vectors for histidine-tagged ubiquitin (His-Ub) together with control siRNA or UBTD1 siRNA. His-Ub crosslinked forms of ASAH1 were purified (IP: His) and the immunoblot of ASAH1 showed ASAH1 ubiquitylation. The immunoblot of ASAH1 (bottom panel) was performed in parallel to verify the amounts of ASAH1 protein engaged in His-Ub purifications. The immunoblot of UBTD1 shows the level of siRNA depletion. (D) Immunoblot and quantification of p-EGFR (Y1068 or Y1086) and p-STAT3. p-STAT3, p-ERK and p-AKT levels were quantified by calculating the ratio between phospho-protein and total-protein, both normalized to loading control signal. Immunoblot of UBTD1 shows the level of siRNA depletion. n ≥ 3 independent experiments; ns = non-significant, **p<0.01, ***p<0.001; ****p<0.0001; (A,D) two-way ANOVA and Bonferroni's multiple comparisons test; data are mean ± s.e.m. The online version of this article includes the following source data for figure 3:

**Source data 1.** Uncropped western blot for *Figure 3*.
**Source data 2.** Row data for *Figure 3*.

that UBTD1 depletion may impair EGFR ubiquitination. To test this possibility, we monitored by proximity ligation assay (PLA) the interaction between ubiquitin and EGFR with or without EGF stimulation (*Figure 4B*). As expected, in control cells, EGF treatment increased EGFR ubiquitination estimated by protein proximity. In UBTD1 knockdown cells, the amount of ubiquitinated EGFR increased similarly to control cells, demonstrating that UBTD1 depletion does not severely impair EGFR ubiquitination. Since the ubiquitination of the EGFR was not modified (*Figure 4—figure supplement 1B*), we then hypothesized that the intracellular trafficking of EGFR may be impaired. We first determined the amount of EGFR at the cell surface. Using an EGF binding assay, we did not detect any difference between control and UBTD1-depleted cells treated or not with EGF (*Figure 4C*). Next, we performed a time-course of EGF/EGFR endocytosis up to their lysosomal degradation in a pulse-chase experiment using a fluorescent-labeled EGF (*Figure 4D*). Cells were stimulated with a fluorescent-labeled EGF, and after washing to remove excess labeled EGF in the medium, the fluorescence detected in the cell corresponds to the EGF endocytosed through its receptor (*Hanafusa et al., 2011*; *Futter et al., 1996*). The amount of internalized EGF remained constant during the first 30 min in both control and UBTD1 knockdown cells, suggesting that the internalization remains similar. In control cells, the amount of EGF started to decay at 120 min, whereas in UBTD1-depleted cells degradation was delayed by at least one hour (*Figure 4D*). To determine in which compartment the EGF/EGFR complex was delayed, we combined a pulse-chase of labeled EGF with immuno-localization of early endosomes (EEA1) or late endosome/lysosome (LAMP1) markers. The arrival and departure of EGF in EEA1-positive compartments was similar in UBTD1-depleted and control cells (*Figure 4E*), confirming that UBTD1 depletion does not alter endocytosis and the first steps of EGFR intracellular trafficking. However, the extent of co-localization between EGF and LAMP1 strongly increased from the 120 min time point in cells depleted for UBTD1 and remained significantly higher for the last two time-points compared to controls (*Figure 4F*). These results underlined that UBTD1 depletion delays the delivery of EGF/EGFR to the degradative lysosomal compartment or impairs the degradative functions of lysosomes.

## UBTD1 depletion slows down EGFR degradation without affecting the overall endolysosomal kinetics

Endocytosis from the plasma membrane to the lysosomes is controlled at multiple stages which makes it virtually impossible to check one by one (*Bakker et al., 2017*). In our proteomic analysis, we identify many endocytosis-associated proteins as potential UBTD1 interactors (*Figure 2—figure supplement 1F*). Therefore, we further considered endocytosis as a global functional flux to evaluate the effect of UBTD1 depletion on its functioning. We first postulate that the delay in EGFR degradation we found in UBTD1-depleted cells can reflect a failure of the whole endolysosomal degradation route. To tackle this possibility, we analysed the impact of UBTD1 knockdown on the different endolysosomal compartment's morphology. During the pulse-chase of EGF, the size and number of EEA1- and LAMP1-positive vesicles was unchanged between control and UBTD1-depleted cells (*Figure 4—figure supplement 1C–F*). Consistent with this, the number and morphology of the early endosomes, late endosomes, and lysosomes were similar at the ultrastructural level (*Figure 4—figure supplement 1G,H*). Although the endocytic compartments were present and morphologically normal, the flux of cargoes along these compartments could be altered. To test it, we loaded the

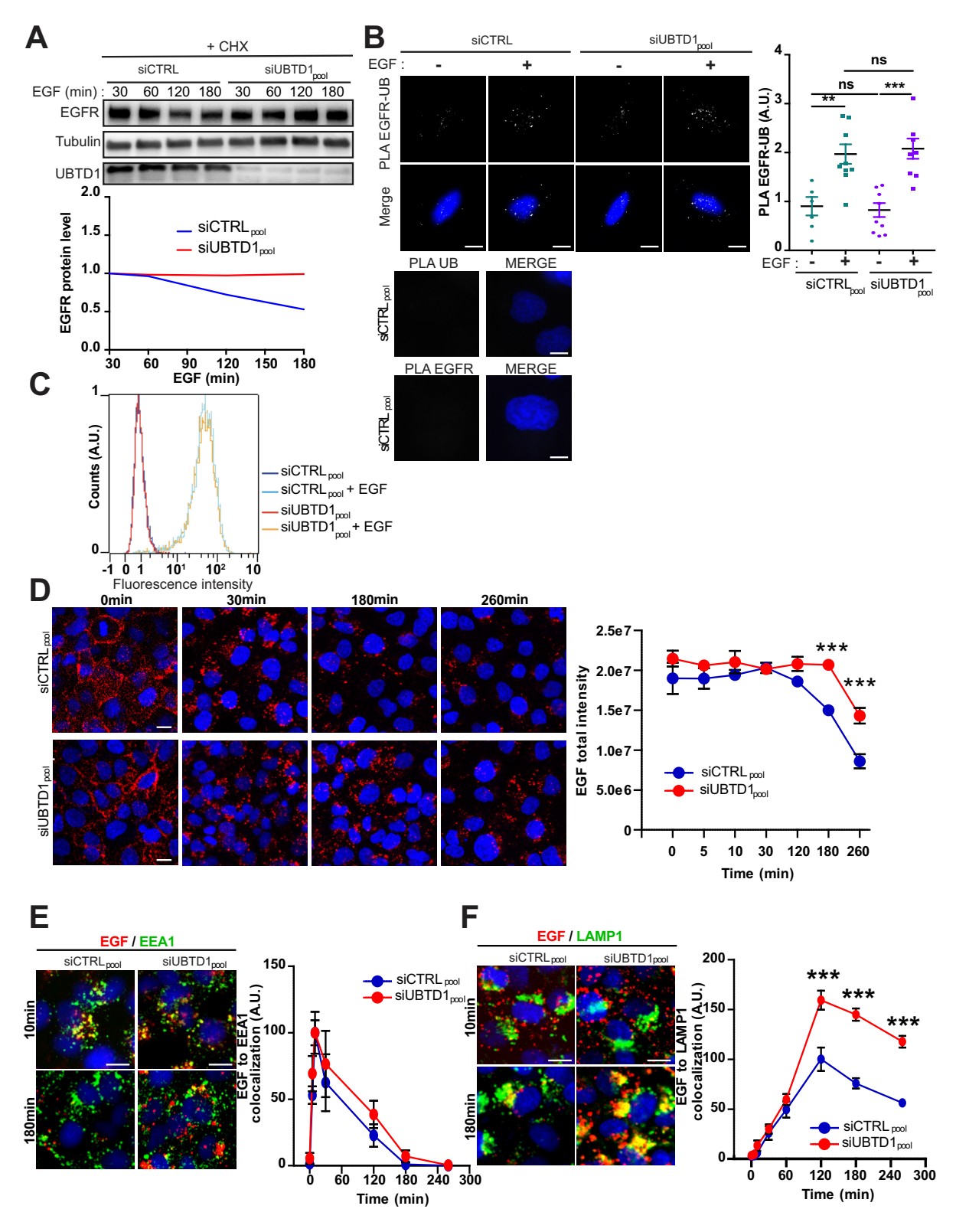

**Figure 4.** UBTD1 depletion slows-down EGFR degradation. (A–F) DU145 cells were transfected for 48 hr with the indicated siRNA (siCTRLpool or siUBTD1pool). (A) Immunoblots (up) and quantification (down) of EGFR levels in cells treated with cycloheximide (CHX) for 2 hr and in presence of EGF (50 ng/ml) at different time points. Immunoblot of UBTD1 shows the level of siRNA depletion. Tubulin was used as a loading control. (B) Proximal ligation assay monitoring and quantification of EGFR associated with ubiquitin in DU145 treated with EGF (50 ng/ml, 10 min). Nuclei were stained with
*Figure 4 continued on next page*

*Figure 4 continued*

DAPI (blue) on the MERGE image. (C) EGF binding to cell surfaces detected by flow cytometry. (D) EGF-alexa647 pulse chase images and quantification. (D) Representative images and quantification of EGF and EEA1 co-localization during EGF-alexa647 pulse chase. (E,F) Representative images and quantification of EGF and LAMP1 co-localization during EGF-Alexa647 pulse chase. Scale bar = 10 µm. n ≥ 3 independent experiments; ns = non-significant; **p<0.01; ***p<0.001; (B, D–F) Bonferroni's multiple comparison test; data are mean ± s.e.m.

The online version of this article includes the following source data and figure supplement(s) for figure 4:

**Source data 1.** Uncropped western blot for *Figure 4*.
**Source data 2.** Row data for *Figure 4* and for *Figure 4—figure supplements 1–2*.
**Figure supplement 1.** UBTD1 depletion slows down EGFR degradation.
**Figure supplement 1—source data 1.** Uncropped western blot for *Figure 4—figure supplement 1*.
**Figure supplement 1—source data 2.** Uncropped western blot for *Figure 4—figure supplement 2*.
**Figure supplement 2.** UBTD1 depletion slows down RTK degradation.

cells with quantum dots coupled to BSA (DQ-BSA), a fluid phase cargo that fluoresces when reaching the lysosomes (*Marwaha and Sharma, 2017*). Both the number of DQ-BSA positive vesicles and the total DQ-BSA intensity were unchanged between control and UBTD1-depleted cells (*Figure 4—figure supplement 1I–K*). Because the lysosomes were accessible to internalized cargoes and were functional (DQ-BSA fluoresces when lysosomes are functional), it is thus likely that the defect in EGFR degradation observed in UBTD1-depleted cells is not due to a general disruption of the endolysosomal degradation pathway, but is rather restricted to a subset of proteins including EGFR. To test whether this effect can be extended to other tyrosine kinase receptors, we performed pulse-chase experiments with a fluorescent-labeled HGF to monitor the temporal kinetics of its degradation in the lysosome. As for the EGF, in UBTD1-depleted cells, we observe a delay in the extinction of the intracellular fluorescent-HGF signal (*Figure 4—figure supplement 2A*). Next, we tested the effect of UBTD1 knockdown on the ubiquitination of c-Met. As with EGFR, we found that UBTD1 depletion does not alter c-Met ubiquitination (*Figure 4—figure supplement 2B,C*). Lastly, we wanted to know if, as for the EGFR, the depletion of UBTD1 affects the total amount of c-Met (*Figure 4—figure supplement 2D*). Indeed, in UBTD1-depleted cells, we observe an increase in the level of c-Met and TGF-β R without any change in their mRNA expression (*Figure 4—figure supplement 2E*). Nevertheless, this mechanism seems to be restricted to tyrosine kinase receptors since we do not observe any change in the amount of IL2 or Transferrin receptors (*Figure 4—figure supplement 2D*). Collectively, this set of data shows that UBTD1 depletion alters degradation of some tyrosine kinase receptors including EGFR without interfering with the morphology or the functionality of the endolysosomal machinery.

## UBTD1 controls p62/SQSTM1 ubiquitination and endolysosomal vesicle positioning

Because the defect in EGFR degradation observed in UBTD1-depleted cells is specific to EGFR rather than a general defect in endocytic trafficking, and because we identify EGFR as a potential UBTD1 partner (*Figure 2—figure supplement 1F*), we decided to focus on UBTD1 and EGFR common interactors. Thus, we re-examined our proteomic data, generated a new protein-protein network centered around EGFR, and applied cell compartment filters to refine the analysis (*Figure 5A, B*). Using this approach, we identified a minimal interaction network between EGFR and p62/SQSTM1 (*Figure 5C*). Strikingly, p62/SQSTM1 was shown to be involved in the positioning of EGFR-positive endolysosome at the ER (*Jongsma et al., 2016*), which is critical for lysosomal function (*Pu et al., 2016*). We first confirmed that p62/SQSTM1 co-immunoprecipitated with UBTD1, demonstrating that these two proteins interact or, at least, are in the same protein complex (*Figure 5D*). Because interactions between endolysosome and the ER rule the spatial positioning of the endolysosomal vesicles, we analysed the pattern distribution of the early endosome (EEA1) and the late endosome/lysosome (LAMP1) vesicles in control and UBTD1-depleted cells. UBTD1 depletion does not affect the distribution of EEA1-positive vesicles nor their co-localization with the ER marker calreticulin (*Figure 5E*, *Figure 5—figure supplement 1A*). In contrast, UBTD1 depletion scatters LAMP1-positive vesicles and decreases the co-localization between LAMP1 and calreticulin (*Figure 5F*, *Figure 5—figure supplement 1B*). Consistent with this, in cells stably overexpressing GFP-tagged UBTD1, the LAMP1-positive vesicles, but not the EEA1-positive vesicles, were more clustered in the

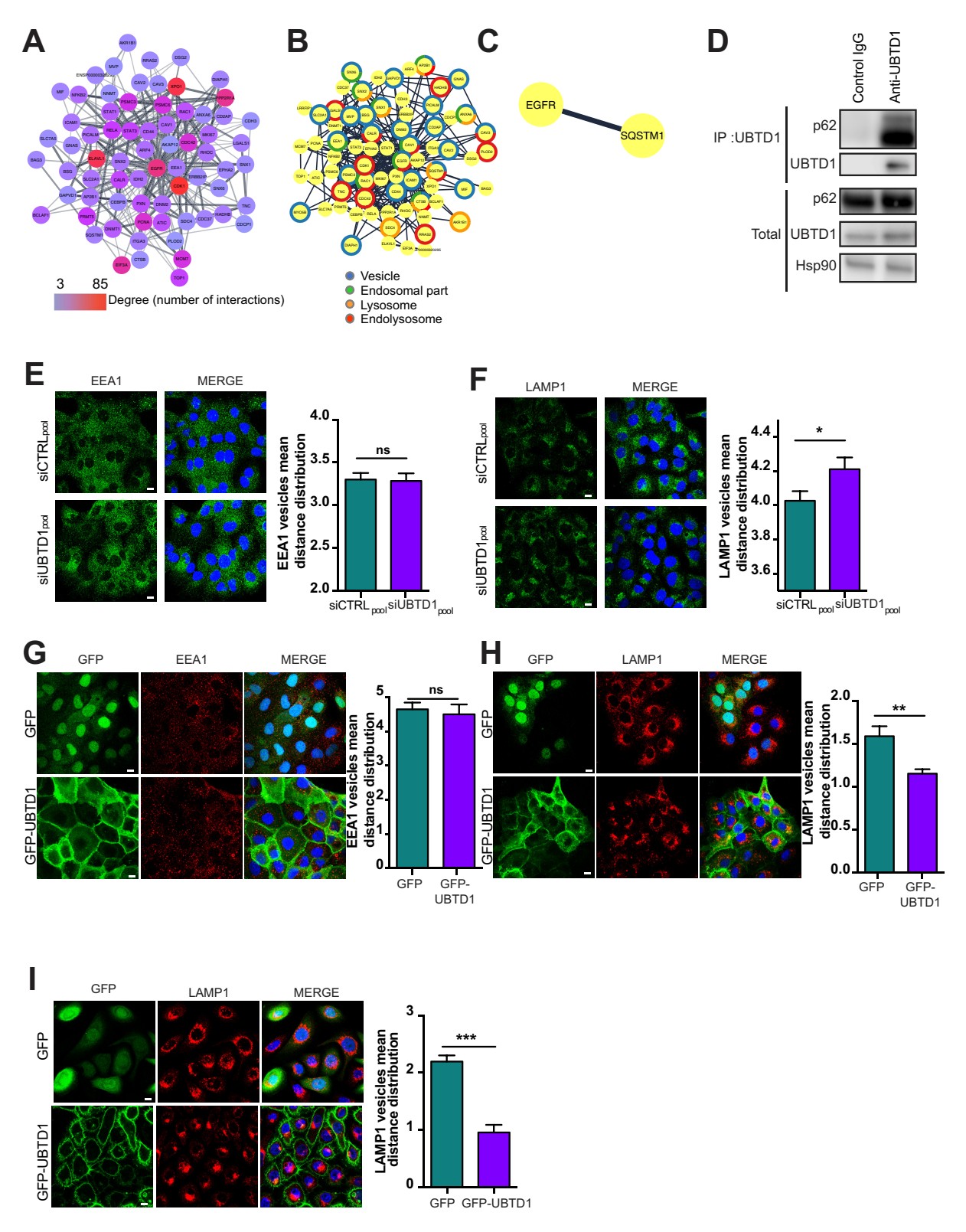

**Figure 5.** UBTD1 interacts with p62/SQSTM1 and controls endolysosomal vesicle positioning. (**A**) UBTD1 protein interactome in DU145 cell line centered on EGFR (first neighbors). Node color scale (from blue to red) illustrates the degree (number of interactors). The edges (connecting lines) represent the interactions between proteins reported in String database. The thickness of the edge represents the interaction score (string database, combined score). (**B**) Go Enrichment analysis (http://geneontology.org). Most relevant Go component are presented by using a split donut color code

*Figure 5 continued on next page*

*Figure 5 continued*

(**C**) Minimal protein interaction network derived from graph (**A**) by applying subcellular localization filters focused on endosome, ER, and lysosome (StringApp, Cytoscape). (**D**) Co-immunoprecipitation in DU145 cells between endogenous p62 and UBTD1. UBTD1 was used as bait. The IgG isotype was used as a negative control. (**E,F**) DU145 cells were transfected for 48 hr with the indicated siRNA (siCTRLpool or siUBTD1pool). Representative confocal immunofluorescence images (left) and quantification (right) of EEA1 (**E**) or LAMP1 (**F**) vesicle distribution. (**G,H**) DU145 or (**I**) RWPE cells were stably transduced with GFP or GFP-UBTD1. Representative confocal immunofluorescence images (left) and quantification (right) of EEA1 (**G**) or LAMP1 (**H,I**) vesicles distribution. Scale bar = 10 µm. n $\geq$ 3 independent experiments; ns = non-significant; *p<0.05, **p<0.01, ***p<0.001; (**E–I**) two-tailed t-test; data are mean ± s.e.m.

The online version of this article includes the following source data and figure supplement(s) for figure 5:

**Source data 1.** Uncropped western blot for *Figure 5*.
**Source data 2.** Row data for *Figure 5* and for *Figure 5—figure supplement 1*.
**Figure supplement 1.** UBTD1 controls p62/SQSTM1 ubiquitination and endolysosomal vesicles positioning.

perinuclear region (*Figure 5G,H*). To validate this result further, we reproduced this experiment in a human epithelial cell line (RWPE1). Again, UBTD1 over-expression lead to a massive clustering of LAMP1-positive vesicles in the perinuclear region (*Figure 5I*). Taken together, both over- and down-expression experiments support a role for UBTD1 in controlling the positioning of the late endolysosomal degradative compartment.

Because p62/SQSTM1 is activated by ubiquitination *via* the ER-resident E3 ligase RNF26 (*Jongsma et al., 2016*), we performed a p62/SQSTM1 ubiquitination assay in UBTD1 depleted cells (*Figure 6A*). As depicted in *Figure 6A*, UBTD1 knockdown drastically reduced p62/SQSTM1 ubiquitination and increases both p62/SQSTM1 level and protein half-life (*Figure 6B,C*, *Figure 6—figure supplement 1A,B*). To confirm that UBTD1 acts on p62/SQSTM1 ubiquitination specifically via RNF26, we examined the interaction between p62/SQSTM1 and RNF26 in control and UBTD1-depleted cells (*Figure 6D*, *Figure 6—figure supplement 1C*). As shown in *Figure 6D*, UBTD1 sharply increases the proximity between P62/SQSTM1 and RNF26 thereby promoting the ubiquitination of P62/SQSTM1 by RNF26. We then verified that RNF26, by modulating cargoes trafficking, contributes to the net sum of EGFR present in the cell. The knockdown of RNF26 increases the amount of EGFR and p62 broadly corroborating the elegant experiments performed by *Jongsma et al., 2016* (*Figure 6E*). Interestingly, in UBTD1-depleted cells, RNF26 depletion does not have an additive effect on EGFR or P62 level, suggesting that UBTD1 and RNF26 act on the same molecular mechanism.

We then wondered whether the role of UBTD1 was related to the function of RNF26 or whether UBTD1 was more broadly associated with the ubiquitination of P62/SQSTM1. For this purpose, we stimulated autophagy in RWPE1 cells and then treated them with bafilomycin to block lysosome acidification (*Figure 6F*, *Figure 6—figure supplement 1D*). Despite an increase in autophagy in the UBTD1-depleted cell, we found that the general autophagic flow is not blocked and the accumulation kinetics of LC3II and P62/SQSTM1 are similar to control cells, confirming that UBTD1 acts on P62/SQSTM1 ubiquitination specifically via RNF26. These data demonstrate that UBTD1 controls late endosome/lysosome positioning and strongly suggest that the interaction of EGFR-positive vesicles with the ER is disrupted by a defect of p62/SQSTM1 ubiquitination by RNF26.

Collectively, we here showed that UBTD1 plays a crucial role in the post-translational regulation of the EGFR (*Figure 6—figure supplement 1E*). On the one hand, by modulating the cellular level of GM3 through ASAH1 ubiquitination, UBTD1 controls the ligand-independent phosphorylation of EGFR. On the other hand, UBTD1, via the ubiquitination of SQSTM1/p62 by RNF26 and positioning of late endosome/lysosome, participates in the lysosomal degradation of EGFR. The coordination of these two ubiquitin-dependent processes contributes to the control of the duration of the EGFR signal and cell proliferation. In conclusion, we have highlighted a yet unsuspected role of UBTD1 on receptor signaling which could be of major importance in certain human pathologies. Moreover, our data lead us to propose that UBTD1 may represent a multistrata coordinator orchestrating EGFR signaling in space and time.

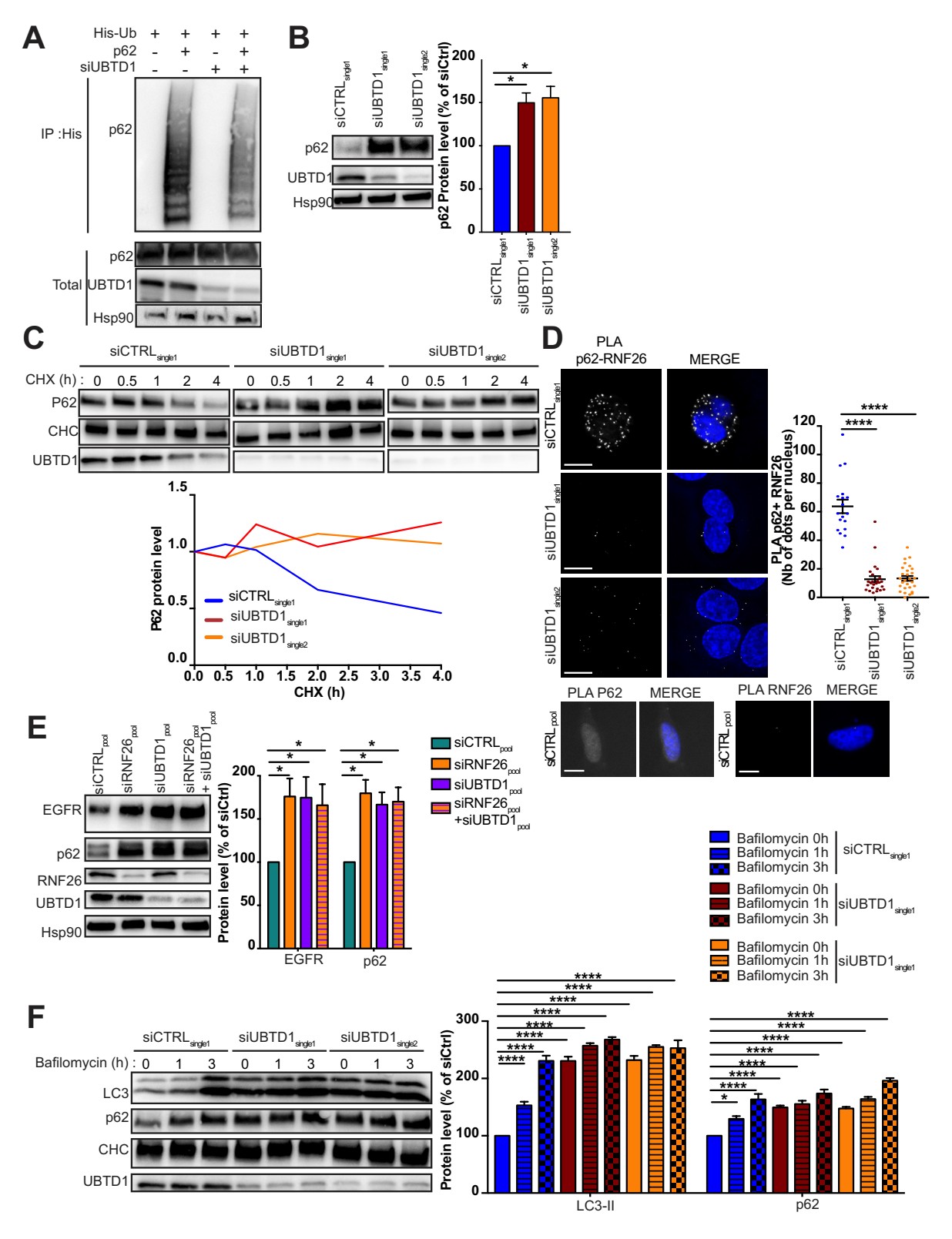

**Figure 6.** UBTD1 controls p62/SQSTM1 ubiquitination. (A–F) DU145 cells were transfected for 48 hr with the indicated siRNA (siCTRLpool or siUBTD1pool or siUBTD1single1 or single2 or RNF26: siRNF26pool). (A) Immunoblots show p62 ubiquitylation in HEK cells in different experimental conditions. Cells were transfected, as indicated, with expression vectors for histidine-tagged ubiquitin (His-Ub) together with control siRNA or UBTD1 siRNA. His-Ub crosslinked forms of p62 were purified (IP: His) and the immunoblot of p62 showed p62 ubiquitylation. The immunoblot of p62 (bottom

*Figure 6 continued on next page*

*Figure 6 continued*

panel) was performed in parallel to verify the amounts of p62 protein engaged in His-Ub purifications. The immunoblot of UBTD1 shows the level of siRNA depletion. (**B**) Immunoblot and quantification of p62. The immunoblot of UBTD1 shows the level of siRNA depletion. (**C**) Immunoblots (up) and quantification (down) of p62 levels in cells treated with cycloheximide (CHX) at different time points. Immunoblot of UBTD1 shows the level of siRNA depletion. CHC was used as a loading control. (**D**) Proximal ligation assay monitoring and quantification of p62 associated with RNF26. (**E**) Immunoblot and quantification of p62 and EGFR. The immunoblots of RNF26 and UBTD1 show the level of siRNA depletion. (**F**) Immunoblot and quantification of LC3 and p62 in cells treated or not with bafilomycin at different time points. The immunoblot of UBTD1 shows the level of siRNA depletion. Scale bar = 10 µm. n ≥ 3 independent experiments; *p<0.05, ****p<0.0001; (**B,D**) Bonferroni's multiple comparison test; (**E–F**) two-way ANOVA and Bonferroni's multiple comparisons test; data are mean ± s.e.m.

The online version of this article includes the following source data and figure supplement(s) for figure 6:

**Source data 1.** Uncropped western blot for *Figure 6*.
**Source data 2.** Raw data for *Figure 6* and for *Figure 6—figure supplement 1*.
**Figure supplement 1.** UBTD1 controls p62/SQSTM1 ubiquitination.
**Figure supplement 1—source data 1.** Uncropped western blot for *Figure 6—figure supplement 1*.

## Discussion

The ubiquitination process works in a hierarchical framework with few E2 enzymes allowing the ubiquitination of many substrates by association with multiple E3 enzymes (*Ye and Rape, 2009*). Previously, we found that UBTD1 interacts with a subset of E2s to form stable stoichiometric complexes (*Uhler et al., 2014*). Based on these biochemical results, it seems very likely that UBTD1, through its interaction with some E2s, can modify the ubiquitination of many proteins. Illustrating this assumption, we previously demonstrated that UBTD1 controls the ubiquitination of yes associated protein (YAP) and similar findings have been reported for MDM2 (mouse double minute two homolog) (*Torrino et al., 2019*; *Zhang et al., 2015*). However, these two specific examples presumably reflect only a small portion of the UBTD1 target proteins and, more importantly, of the cellular processes regulated by UBTD1. By using a holistic approach, we here provide evidence that UBTD1 coordinates the EGFR signaling pathway by controlling two distinct ubiquitin-mediated mechanisms. First, UBTD1 limits the self-phosphorylation of EGFR by modulating the ceramide (GM3) balance through the ubiquitin-dependent degradation of the acid ceramidase ASAH1. Additionally, UBTD1 controls the lysosomal degradation of EGFR by adjusting the spatial patterning of late endosome/lysosomes through RNF26-dependent ubiquitination of p62/SQSTM1. Together, the deregulation of these two molecular checkpoints causes EGFR self-phosphorylation and increases intracellular EGFR lifetime which leads to the persistence of its signaling and induces cell proliferation.

In UBTD1-depleted cells, the membrane lipid composition was altered by a decrease in ASAH1 ubiquitination. ASAH1 is synthesized in the ER as an inactive proenzyme and must be activated through autocleavage to become active in the lysosome (*Brannigan et al., 1995*; *Gebai et al., 2018*). This process is fundamental since ASAH1 is prominently involved in a genetic lysosomal storage disorder in human (Farber's disease) (*Schuchman, 2016*). However, the regulatory mechanism remains unclear. Interestingly, it has been reported that ASAH1 is regulated by the proteasome and that the deubiquitinase USP2 modulates protein half-life, clearly suggesting that, ASAH1 is controlled by a ubiquitination-mediated process before reaching the lysosome (*Mizutani et al., 2015*). Our findings reinforce this proposal by showing that UBTD1 acts on the ubiquitination of ASAH1. Furthermore, ASAH1 has been described by others as potentially interacting with the E3 ligase HERC2, which we also identified in our proteomic screen as a putative UBTD1 partner (*Galligan et al., 2015*). These data lead us to speculate that the lysosomal content of ASAH1 could be regulated by a ubiquitination process mediated by UBTD1 and HERC2. This hypothesis, although interesting, is still speculative and needs to be further explored in the future. Interestingly, Simmons' group nicely demonstrated that the ceramide GM3 prevents EGFR self-phosphorylation (*Coskun et al., 2011*). In accordance with this work, we observed that UBTD1 depletion, by decreasing cellular GM3 content, increases EGFR self-phosphorylation. In a broader view, we propose that UBTD1 regulates EGFR self-phosphorylation by controlling GM3 content through ASAH1 ubiquitination.

In a genome-wide screen analysis performed to identify endocytic trafficking master regulators, it has been reported that UBTD1 knockdown increases total EGF vesicle intensity, without major changes in vesicle number, elongation, area, or distance to the nucleus (*Collinet et al., 2010*).

Consistent with this finding, we here demonstrate that UBTD1 depletion drastically increases intracellular fluorescent-labeled EGF lifetime without altering the morphology or the distribution of the early endocytic vesicles. To discriminate at which stage EGF trafficking is corrupted, we analysed EGF kinetics throughout its endolysosomal trafficking. These functional experiments clearly emphasize that UBTD1 depletion only affects the late endosome/lysosome step of EGF transport while totally preserving the upstream EGF flux. As elegantly demonstrated by *Jongsma et al., 2016*, SQSTM1/P62 is necessary to control EGFR-positive vesicle spatial positioning. Briefly, ubiquitinated p62/SQSTM1 captures specific endolysosomal vesicle adaptors to permit transient ER/endolysosomal contacts which are required for trafficking of some cargoes including EGFR (*Jongsma et al., 2016*). This sophisticated process is orchestrated by the ubiquitination of p62/SQSTM1 by the ER-resident-E3 ligase, RNF26, and requires spatial and temporal coordination. Hence, we propose that UBTD1 impairs EGFR-positive vesicle positioning through a p62/SQSTM1-RNF26 mechanism close to the one described by *Jongsma et al., 2016*. Importantly, this process is presumably not restricted to EGFR since we observe, as anticipated by *Jongsma et al., 2016*, similar effects of UBTD1 depletion on other signaling receptors that use the same intracellular routing as EGFR such as c-Met or TGF-β R (*Park and Richardson, 2020*; *Chen, 2009*). Although UBTD1 depletion affects p62/SQSTM1 ubiquitination by RNF26, it has a relatively minor impact on P62/SQSTM1-dependent autophagy demonstrating a high specificity toward some E3s rather than for the substrate in the ubiquitination process. We also noticed this E3-specificity of UBTD1 for the ubiquitination EGFR. Once the canonical clathrin-dependent EGFR endocytosis is saturated by an excess of EGF, EGFR is rapidly routed toward lysosomal degradation through clathrin-independent endocytosis (*Bakker et al., 2017*; *Todeschini et al., 2008*). This critical switch is controlled by the ubiquitination of EGFR by the E3 ligase c-Cbl and the E2 enzyme UBE2D3 (*Pu et al., 2016*; *Brannigan et al., 1995*). The depletion of UBTD1 had no effect either on the ubiquitination of EGFR or, indeed, on overall EGFR internalization suggesting that UBTD1 is not mandatory for UBE2D3-c-Cbl-mediated EGFR ubiquitination. Although UBTD1 is a molecular partner of some E2s (*Uhler et al., 2014*), it is likely that its function is specific to some E3s or context-dependent.

From the few works that have been published on UBTD1 and despite still incomplete knowledge about its cellular functions, it seems that a global picture is emerging. Although still largely speculative, we propose that UBTD1 is a rather stable partner of some E2s and its function could be to foster, in certain circumstances, the association between E2 and E3 within the ubiquitination complex. This scaffolding function, controlled by a still unknown mechanism, allows to finely control the ubiquitination of some substrates like P62/SQSTM1 or ASAH1.

EGFR plays a major role in cell proliferation and is frequently overexpressed or hyperactivated in many epithelial cancer cells (*Yarden and Pines, 2012*). Considering the dual role of UBTD1 on EGFR autophosphorylation and lysosomal degradation, it is not surprising that UBTD1 depletion exacerbates EGFR signaling and induces an EGFR-dependent cellular proliferation. This last finding underlines the importance of UBTD1 in EGFR-driven cell proliferation and may be further investigated in an EGFR hyper-activated state such as cancer. Collectively, we here demonstrated that UBTD1 acts as a coordinator of EGFR signaling, illustrating that the regulation of E2/E3 enzymes of the ubiquitination system by scaffold proteins may represents a critical but still underestimated control layer for coordination of some signaling pathways.

## Materials and methods

### Reagents and antibodies

EGF (50 ng/ml, AF-100–15) or 10 ng/ml (cell proliferation assay) and HGF (40 ng/ml, 100–39H) were purchased from PeproTech. Anti-UBTD1 (HPA034825, RRID:AB_10602254), sodium chloride, DAPI, Gefitinib (0,5 µM) (SML1657), cycloheximide (75 µM), bafilomycinA1 (100 nM), and 2,5-dihydroxybenzoic acid (DHB) were purchased from Sigma-Aldrich. Anti-HSP90 (sc-13119, RRID:AB_675659), anti-Tubulin (sc-398937), anti-RhoGDI (sc-360, RRID:AB_2227516), and anti-Ubiquitin (sc-8017, RRID:AB_2762364) were purchased from Santa Cruz Biotechnology. Anti-p-STAT3 (Tyr705) (clone D3A7, 9145, RRID:AB_2491009), anti-STAT3 (clone 124H6, 9139, RRID:AB_331757), anti-ERK (#9102, RRID:AB_330744), anti-p-ERK (Thr202/Tyr204, #4370, RRID:AB_2315112), anti-Akt (4691, RRID:AB_915783), anti-p-Akt (Ser473) (#4060, RRID:AB_2315049), anti-p-Akt (Thr308) (#13038), anti-c-Met

(#4560), anti-TGF-β Receptor II (#79424, RRID:AB_2799933), anti-p-EGFR Tyr1068 (#3777, RRID:AB_2096270), anti-p-EGFR Tyr1086 (2234, RRID:AB_331701), anti-LC3B (2775, RRID:AB_915950), and anti-EGFR (2232, RRID:AB_331707) were purchased from Cell Signaling Technology. Anti-EEA1 (610457, RRID:AB_397830), anti-Clathrin Heavy Chain (610499) and anti-CD107a/LAMP1 (555798, RRID:AB_396132) were purchased from BD Biosciences. Anti-SQSTM1/P62 (GTX100685, RRID:AB_2038029) was purchased from Genetex, anti-ASAH1 (NBP1-89296, RRID:AB_11025126) and anti-CerS2/LASS2 (NBP1-84537, RRID:AB_11033791) from Novus Biologicals. Anti-calreticulin-ER Marker (ab2907, RRID:AB_303402) and anti-RNF26 (ab236791) were purchased from Abcam. Anti-IL-2R (MAB623, RRID:AB_2125599) was purchased from R and D Systems. DQ Red BSA (D12051) and anti-Transferrin Receptor (# 13–6800, RRID:AB_2533029) were purchased from Invitrogen. Ganglioside GM3 (860058) was purchased from Avanti Polar Lipids. HRP-conjugated donkey anti-mouse IgG (715-035-150) and HRP-conjugated anti-mouse IgG (711-035-152) were purchased from Jackson ImmunoResearch Laboratories. Epidermal Growth Factor, biotinylated, complexed to Alexa Fluor 647 Streptavidin (Alexa Fluor 647 EGF complex) was purchased from Molecular Probes (Invitrogen, E35351). HGF was complexed to Alexa Fluor 647 (Labeling Kit) according to provider's protocol (A30009, Molecular Probes, Invitrogen).

## Cell culture

DU145 cells were purchased from the American Type Culture Collection (ATCC, RRID:CVCL_0105). All cells used in this study were within 20 passages after thawing. DU145 cells were cultured (37°C, 5% $CO_2$) in Dulbecco's modified Eagle's medium (DMEM, Gibco) supplemented with 10% fetal bovine serum (Gibco) and penicillin/streptomycin (1%, Gibco). The RWPE-1 cell line was obtained from ATCC (CRL-11609; RRID:CVCL_3791). RWPE-1 cells were maintained in KSFM (Life Technologies) supplemented with 5 ng/mL epidermal growth factor (Life Technologies), 50 mg/mL bovine pituitary extract (Life Technologies), and 1% penicillin-streptomycin (Life Technologies). The cells were routinely cultured in a humidified atmosphere with 5% $CO_2$ at 37°C. All cell lines were found to be *Mycoplasma* free.

## siRNA and plasmid transfection

Transfections were performed with Lipofectamine RNAiMAX according to the manufacturer's instructions (Invitrogen) using siRNA SMARTpool or individual ON-TARGETplus (Horizon Discovery Ltd) Human UBTD1 (L-018385-00-0005), Human ASAH1 (L-005228-03-0005, J-005228-05-0002, J-005228-06-0002), Human RNF26 (L-007060-00-0005), or ON-TARGETplus Non-Targeting Control (D-001830–10, D-001810–01).

Sequences of single siRNA UBTD1Human have been synthesized by Eurogenetec using the following sequences: UBTD1single1 sense CAAGCGAGCAGGACGCAAU/antisense GGAGCAAACGGGAUGAGUU; UBTD1single2 sense GAAGCAGGUUCGAGCCAC/antisense CCACAAGGGCCAACCAGGA. Lipofectamine 2000 was used for plasmid transfection according to the manufacturer's instructions (11668, Invitrogen). Histidine-tagged ubiquitin (pCI-His-hUbi, #31815), EGFR-GFP (#32751), c-MET-GFP (#37560) and Ha-tagged p62 (HA-p62, #28027) plasmids were purchased from Addgene. ASAH1 human plasmid (RC212434) was purchased from Origene technologies. DNA constructs corresponding to the mature form of human UBTD1 were subcloned from pEGFP-N1 (Novagen) to a retroviral vector compatible plasmid (PPRIG) (*Albagli-Curiel et al., 2007*). Cells were transduced with an MLV-based retroviral vector and selected by puromycin.

## Human phospho-kinase antibody array

Relative phosphorylation levels of 43 kinases and two related proteins were assessed using the Proteome Profiler Human Phospho-Kinase Array Kit (R and D Systems), according to the manufacturer's instructions. In brief, cell lysates were incubated overnight with nitrocellulose membranes of the Human Phospho-Kinase Array (R and D Systems). Membranes were then washed, incubated with biotinylated detection antibody cocktails, and then incubated with streptavidin-horseradish peroxidase and visualized using ECL (Millipore) and analysed on Pxi (Syngene). The signal of each capture spot was measured using the 'Protein Array Analyzer for ImageJ' and normalized to internal reference controls. Heatmaps were generated using Graphpad Prism software.

## Cell proliferation assay

After 48 hr of siRNA transfection, cells were seeded at $5 \times 10^4$ cells/well in a 96-well plate and cultured for 24 hr prior to experiment. All conditions were performed in triplicate. Cells transfected with control siRNA served as control. Proliferation was measured using the Cell Proliferation ELISA BrdU kit (Roche Diagnostics GmbH), according to the manufacturer's instructions. Briefly, cells were labeled with BrdU at a final concentration of 10 µM/well, for 12 hr at 37°C. The cells were then denatured with FixDenat solution and incubated for 120 min with 1:100 diluted mouse anti-BrdU conjugated to peroxidase. After two washings (PBS $1\times$), the substrate solution was added for 25 min and, after this period, the reaction was stopped with 1 M H2SO4 solution. Absorbance was measured within 5 min at 450 nm with a reference wavelength at 690 nm using an ELISA plate reader.

## Kinetic growth assay

Cells were transduced with NucLight Red Lentivirus and selected with puromycin (1 µg/ml) for 2 weeks (Essen Biosciences). After siRNA transfection, cells were allowed to grow (12-well plates) for 48 hr under a live cell imaging system (Essen Biosciences). The experiments were carried out three times independently. Nine images per well (six wells per condition) were taken every 2 hr for 48 hr and analysed with the Incucyte analysis software. The proliferation rate was calculated with the slope between 0 and 48 hr.

## Immunofluorescence and pulse-chase

For immunofluorescence analysis, the cells were fixed with PBS/PFA 4% for 10 min and permeabilized with PBS/Triton 100 $\times$ 0.2% for 5 min. After blocking with PBS/BSA 0.2% for 1 hr, the cells were then incubated with primary antibodies (1/100) at room temperature for 1 hr. Secondary antibodies coupled with Alexa-594 and/or Alexa-488 (A-11012, A-11001, Life technologies) were used at 1/500 for 1 hr. Nuclei were counterstained with DAPI (Sigma-Aldrich). For the pulse-chase experiments, the cells were incubated for 10 min with 100 ng/ml of EGF-alexa647 or 40 ng/ml of HGF-Alexa647 with at 4°C and chased with fresh media for 0, 5, 10, 30, 60, 120, 180, and 240 min at 37°C (*Collinet et al., 2010*). Then, the cells were washed, fixed, and immunolabeled as described above. Images were obtained in a randomized fashion using an LSM510 confocal microscope (Zeiss) and a Nikon A1R confocal. At least 20 images per condition were analysed using the unbiased multiparametrics MotionTracking software, as previously described (generous gift from Dr Y. Kalaidzidis, Marino Zerial's lab) (*Gilleron et al., 2013*; *Zeigerer et al., 2012*).

## Electron microscopy

For ultrastructural analysis, cells were fixed in 1.6% glutaraldehyde in 0.1 M phosphate buffer, rinsed in 0.1 M cacodylate buffer, post-fixed for 1 hr in 1% osmium tetroxide and 1% potassium ferrocyanide in 0.1 M cacodylate buffer to enhance the staining of membranes. Cells were rinsed in distilled water, dehydrated in alcohols and lastly embedded in epoxy resin. Contrasted ultrathin sections (70 nm) were analysed under a JEOL 1400 transmission electron microscope mounted with a Morada Olympus CCD camera. For quantification, at least 200 structures were counted from at least 20 images per condition acquired randomly.

## Proximity Ligation Assay

The proximity ligation assay (PLA) kit was purchased from Sigma-Aldrich, and the assay was performed according to the manufacturer's protocol. Cells were fixed in 4% PFA for 10 min at room temperature, quenched in 50 mM NH4Cl for 10 min, permeabilised with 0.2% Triton X-100 (w/v) for 10 min and blocked in PBS/BSA. Cells were incubated with primary antibodies for 45 min in PBS/BSA. Coverslips were mounted in Fluoromount with DAPI to stain nuclei. PLA signals were visible as fluorescent dots and imaged using an inverted epifluorescence Leica DM 6000B microscope equipped with an HCX PL Apo 63x NA 1.32 oil immersion objective and an EMCCD camera (Photometrics CoolSNAP HQ). Fluorescent dots were quantified using ImageJ. Cells and nuclei were delineated to create masks. After a Max Entropy threshold, the PLA dots were quantified in both masks with the ImageJ Analyze Particles plugin. All counts were divided by the number of cells.

## Flow cytometric analysis

After 48 hr of siRNA transfection, expression of cell surface EGFR was monitored by flow cytometry on living cells without permeabilization. After washing in ice-cold PBS, cells were immunolabeled on ice for 10 min with Alexa Fluor 647 EGF complex (Molecular Probes, Invitrogen, E35351) in PBS containing 2% BSA. After two further washes in ice-cold PBS containing 2% BSA and one wash in ice-cold PBS, cells were suspended in PBS and analysed by flow cytometry on a FACS (MACSQuant VYB, Miltenyi Biotec); data were analysed using the Cell Quest software. Measurements were compared to the isotopic control (APC-conjugated anti-mouse IgG1, Biolegend clone MOPC-21 #BLE400122, 1/100, Ozyme) to determine background and positivity thresholds. Each experiment was repeated at least three times.

## Western blot assays

Cells were lysed in RIPA buffer (Pierce) or directly in Laemmli's buffer. After denaturation, protein lysates were resolved by SDS-PAGE and transferred onto PVDF membranes (Millipore). Membranes were blocked with 2% BSA in TBS tween20 0.1% and incubated in the presence of the primary and then secondary antibodies. After washing, immunoreactive bands were visualized with ECL (Millipore) and analysed on Pxi (Syngene).

## Ubiquitylation assays

HEK293 cells ($5 \times 10^6$) were transfected with 10 µg of His-tagged ubiquitin WT expression vectors together with p62, ASAH1, EGFR or MET and siRNA targeting UBTD1. Ubiquitylated proteins were recovered by His-tag affinity purification on cobalt resin in urea denaturing conditions, as described (*Torrino et al., 2011*).

## Immunoprecipitation

For endogenous immunoprecipitation, cells were harvested and lysed in NP-40 buffer containing a protease inhibitor cocktail (Thermo Fisher). The lysates (13,200 rpm, 10 min, 4°C) were incubated for 2 hr with the indicated antibody (IP) at 4°C. Then, 10 µl Dynabeads protein A (Invitrogen) was added to each aliquot for 45 min at 4°C. Beads were washed 3–5 times with NP-40 and eluted by boiling in 2x sample buffer at 95°C for 10 min. The eluted fractions were analysed by western blot.

## RNA isolation and RT-PCR from cell lines

Total RNA was extracted by TRIzol reagent according to the manufacturer's instructions (Invitrogen). RNA quantity and quality were determined using NanoDrop One Spectrophotometer (Thermo Scientific). One microgram of total RNA was reverse transcribed into cDNA (A3500, Promega). Real-time quantitative PCR was performed using Fast SYBR Green master mix (Applied Biosystems) on a StepOnePlus System (Applied Biosystems). The gene-specific primer sets were used at a final concentration of 1 µM in a 10 µl final volume. RPLP0 mRNA levels were used as an endogenous control to normalise relative expression values of each target gene. The relative expression was calculated by the comparative Ct method. All real-time RT-PCR assays were performed in triplicate with three independent experiments. Primers are provided below. Primers were published in http://pga.mgh.harvard.edu/primerbank/. EGFR (Forward: AGGCACGAGTAACAAGCTCAC; Reverse:ATGAGGACATAACCAGCCACC); BCL2 (Forward: CGCCCTGTGGATGACT; Reverse: GGGCCGTACAGTTCCA); MMP2 (Forward: TGAGCTATGGACCTTGGGAGAA; Reverse: CCATCGGCGTTCCCATAC); MMP9 (Forward: GAACCAATCTCACCGACAGG; Reverse: GCCACCCGAGTGTAACCATA); HIF2 (Forward: TTGCTCTGAAAACGAGTCCGA; Reverse: GGTCACCACGGCAATGAAAC); UBTD1 (Forward: GCGGTGACAGGCAGTAGAT; Reverse: CGGAGCAAACGGGATGAGTT). For mRNA expression of Epigen (Hs02385424), Betacellulin (Hs01101201), Epiregulin (Hs00914313), TGF-α (Hs00608187), Amphiregulin (Hs00950669), HB-EGF (Hs00181813), MET (Hs01565584), IL2R (Hs00174759), Transferin Receptor (Hs00951083), TGFR-II (Hs00234253), and UBTD1 (Hs00227913), Taqman probes have been used, and experiments were performed as recommended by the provider (Applied Biosystems).

## Enzyme-linked immunosorbent assay

The levels of Epigen (CSB-EL007719HU, Clinisciences), Betacellulin (ab189575, Abcam), Amphiregulin (ab222504, Abcam), Epiregulin (BGK14944, Peprotech), TGF-α (BGK 01135, Peprotech), and HB-EGF (BGK 99075, Peprotech) were measured according to the provider protocol. Values are normalized to the total amount of protein.

## In-gel digestion

Protein spots were manually excised from the gel and distained by adding 100 µL of H2O/ACN (1/1). After 10 min incubation with vortexing the liquid was discarded. This procedure was repeated two times. Gel pieces were then rinsed (15 min) with acetonitrile and dried under vacuum. Each excised spot was reduced with 50 µL of 10 mM dithiothreitol and incubated for 30 min at 56°C. Alkylation was performed with 15 µL of 55 mM iodoacetamide for 15 min at room temperature in the dark. Gel pieces were washed by adding successively (i) 100 µL of H2O/ACN (1/1), repeated two times and (ii) 100 µL of acetonitrile. Next, gel pieces were reswelled in 60 µL of 50 mM $NH_4HCO_3$ buffer containing 10 ng/µL of trypsin (modified porcine trypsin sequence grade, Promega) incubated for one hour at 4°C. Then the solution was removed and replaced by 60 µL of 50 mM $NH_4HCO_3$ buffer (without trypsin) and incubated overnight at 37°C. Tryptic peptides were isolated by extraction with (i) 60 µL of 1% AF (acid formic) in water (10 min at RT) and (ii) 60 µL acetonitrile (10 min at RT). Peptide extracts were pooled, concentrated under vacuum and solubilised in 15 µL of aqueous 0.1% formic acid and then injected.

## NanoHPLC-Q-exactive *plus* analysis

Peptide separation was carried out using a nanoHPLC (ultimate *3000*, Thermo Fisher Scientific). 5 µl of peptides solution was injected and concentrated on a µ-Precolumn Cartridge Acclaim PepMap 100 $C_{18}$ (i.d. 5 mm, 5 µm, 100 Å, Thermo Fisher Scientific) at a flow rate of 10 µL/min and using solvent containing H2O/ACN/FA 98%/2%/0.1%. Next peptide separation was performed on a 75 µm i.d. x 500 mm (3 µm, 100 Å) Acclaim PepMap 100 $C_{18}$ column (Thermo Fisher Scientific) at a flow rate of 200 nL/min. Solvent systems were: (A) 100% water, 0.1%FA, (B) 100% acetonitrile, 0.08% FA. The following gradient was used t = 0 min 4% B; t = 3 min 4%B; t = 170 min, 35% B; t = 172 min, 90% B; t = 180 min 90% B (temperature was regulated at 35°C). The nanoHPLC was coupled via a nanoelectrospray ionization source to the Hybrid Quadrupole-Orbitrap High Resolution Mass Spectrometer (Thermo Fisher Scientific). MS spectra were acquired at a resolution of 70 000 (200 m/z) in a mass range of 150–1800 m/z with an AGC target 5e5 value of and a maximum injection time of 50 ms. The 10 most intense precursor ions were selected and isolated with a window of 2 m/z and fragmented by HCD (Higher energy C-Trap Dissociation) with a normalised collision energy (NCE) of 27. MS/MS spectra were acquired in the ion trap with an AGC target 2e5 value, the resolution was set at 17,500 at 200 m/z combined with an injection time of 100 ms. Data were reprocessed using Proteome Discoverer 2.2 equipped with Sequest HT. Files were searched against the Swissprot *Homo sapiens* FASTA database update on September 2018. A mass accuracy of ± 10 ppm was used for precursor ions and 0.02 Da for product ions. Enzyme specificity was fixed to trypsin with two missed cleavages allowed. Because of previous chemical modification, carbamidomethylation of cysteines was set as a fixed modification and only oxidation of methionine was considered as dynamic modification. Reverses decoy databases were included for all searches to estimate false discovery rates and filtered using the Percolator algorithm with a 1% FDR.

A Protein-Protein Interactions Network (PPI-Net) was created using stringApp in Cytoscape (*Szklarczyk et al., 2011*), and proteins have been clustered using the Markov Cluster Algorithm (clusterMaker2, Cytoscape).

## Lipid extraction and analysis

Extraction was performed using 1.5 mL solvent-resistant plastic Eppendorf tubes and 5 mL glass hemolyse tubes to avoid contamination. Methanol, chloroform and water were each cooled down on wet ice before the lipid extraction. Lipids were extracted according to a modified Bligh and Dyer protocol. The cell pellet was collected in a 1.5 mL Eppendorf tube and 200 µL of water was added. After vortexing (30 s), the sample was transferred to a glass tube containing 500 µL of methanol and 250 µL of chloroform. The mixture was vortexed for 30 s and centrifuged (2500 rpm, 4°C, 10 min).

After centrifugation, 300 µL of the organic phase was collected in a new glass tube and dried under a stream of nitrogen. The dried extract was resuspended in 60 µL of methanol/chloroform 1:1 (v/v) and transferred in an injection vial before liquid chromatography and mass spectrometry analysis. Lipid extraction from the membrane purification was carried out with the same protocol in which the solvent volumes used were divided by two. Reverse phase liquid chromatography was selected for separation with a UPLC system (Ultimate 3000, ThermoFisher). Lipid extracts from cells were separated on an Accucore C18 150 × 2.1, 2.5 µm column (ThermoFisher) operated at 400 µl/min flow rate. The injection volume was 3 µL of diluted lipid extract. Eluent solutions were ACN/H2O 50/50 (V/V) containing 10 mM ammonium formate and 0.1% formic acid (solvent A) and IPA/ACN/H2O 88/10/2 (V/V) containing 2 mM ammonium formate and 0.02% formic acid (solvent B). The step gradient used for elution was: 0 min 35% B, 0.0–4.0 min 35% to 60% B, 4.0–8.0 min 60% to 70% B, 8.0–16.0 min 70% to 85% B, 16.0–25 min 85% to 97% B, 25–25.1 min 97% to 100% B, 25.1–31 min 100% B and finally the column was reconditioned at 35% B for 4 min. The UPLC system was coupled to a Q-exactive orbitrap mass spectrometer (ThermoFisher, CA); equipped with a heated electrospray ionization (HESI) probe. This spectrometer was controlled by the Xcalibur software and was operated in electrospray positive mode. MS spectra were acquired at a resolution of 70,000 (200 m/z) in a mass range of 250–1200 m/z. The 15 most intense precursor ions were selected and isolated with a window of 1 m/z and fragmented by HCD (higher energy C-trap dissociation) with normalized collision energy (NCE) of 25 and 30 eV. MS/MS spectra were acquired with the resolution set at 35 000 at 200 m/z. Data were reprocessed using Lipid Search 4.1.16 (ThermoFisher). In this study, the product search mode was used, and the identification was based on the accurate mass of precursor ions and the MS2 spectral pattern. Mass tolerance for precursors and fragments was set to 5 ppm and eight ppm respectively. The m-score threshold was selected at five and the ID quality filter was fixed at grades A, B, and C. [M + H]+, [M + Na]+, and [M + NH4]+ adducts were searched.

## Mass spectrometric analysis for ganglioside

Ganglioside extraction was performed as described with minor modifications (*Lee et al., 2011*). The aqueous upper layers from two extractions (with a mixture of water–methanol–chloroform, W:M:C = 2:2:1) were collected. Two volumes of water were added to precipitate polyglycoceramides. The dried pellet was resuspended with methanol/water (M:W = 1:1) and analysed in reflector-positive and -negative modes on MALDI-TOF/TOF mass spectrometer Ultraflex III (Bruker Daltonics, Bremen, Germany) from 700 to 2500 Da. External calibration was performed by spotting peptide calibration standard II (BrukerDaltonics). Each sample was spotted in triplicate and mixed with DHB matrix on a steel target plate. All mass spectra were generated by summing 1000 laser shots for reflectron ion mode, and 1000 laser shots for the parent mass. Laser power was adjusted between 15% and 30% of its maximal intensity, using a 200 Hz smartbeam laser. MS spectra were acquired in the reflectron ion mode within a mass range from 500 to 2500 Da. Reflectron ion mode was chosen to obtain high detection sensitivity and resolution. FlexAnalysis version 3.0 and updated 3.4 provided by the manufacturer were applied for data processing. The Human Metabolome Database or HMDB 4.0 (https://hmdb.ca/) was used for peak identifications. Heat map and differential intensity analysis with Limma were done using Phantasus (https://artyomovlab.wustl.edu/phantasus/).

## Statistical analysis

All analyses were performed using Prism 6.0 software (GraphPad Inc). A two-tailed t-test was used if comparing only two conditions. For comparing more than two conditions, one-way ANOVA was used with: Bonferroni's multiple comparison test or Dunnett's multiple comparison test (if comparing all conditions to the control condition). Significance of mean comparison is marked on the graphs by asterisks. Error bars denote SEM.

## Acknowledgements

This work was supported by INSERM, the Côte d'Azur University, and by grants from the French National Research Agency (ANR) through the Investments for the program UCA JEDI to SC (ANR-15-IDEX-01), the Young Investigator Program to JG (ANR18-CE14-0035-01-GILLERON) and the ITMO, Plan Cancer. ST was supported by the 'Fondation de France' and TB by the French National Research Agency (ANR-18-CE14-0025). We thank the GIS-IBISA multi-sites platform 'Microscopie

Imagerie Côte d'Azur' (MICA), funded by the 'Conseil Départemental 06' support from ITMO Cancer Aviesan (National Alliance for Lifr Science and Health) within the framework of Cancer Plan. FB is a CNRS investigator. We would like to thank Marino Zerial's lab for giving us access to the MotionTraffic platform for image quantification.

## Additional information

### Funding

| Funder | Grant reference number | Author |
|---|---|---|
| Agence Nationale de la Recherche | ANR-15-IDEX-01 | Stephan Clavel |
| Agence Nationale de la Recherche | ANR18-CE14-0035-01-GILLERON | Jerome Gilleron |
| Fondation de France | | Stéphanie Torrino |
| Agence Nationale de la Recherche | ANR-18-CE14-0025 | Thomas Bertero |

The funders had no role in study design, data collection and interpretation, or the decision to submit the work for publication.

### Author contributions

Stéphanie Torrino, Conceptualization, Formal analysis, Supervision, Validation, Investigation, Visualization, Methodology, Writing - original draft, Writing - review and editing; Victor Tiroille, Maeva Dufies, Charlotte Hinault, Sonia Dagnino, Lucile Fleuriot, Delphine Debayle, Formal analysis; Bastien Dolfi, Anne-sophie Gay, Data curation, Formal analysis; Laurent Bonesso, Data curation; Jennifer Uhler, Writing - review and editing; Marie Irondelle, Software; Sandra Lacas-Gervais, Formal analysis, Visualization; Mireille Cormont, Thomas Bertero, Resources, Writing - review and editing; Frederic Bost, Jerome Gilleron, Supervision, Conceptualization, Resources, Data curation, Software, Formal analysis, Funding acquisition, Investigation, Writing - original draft, Writing - review and editing; Stephan Clavel, Conceptualization, Resources, Data curation, Software, Formal analysis, Supervision, Funding acquisition, Validation, Investigation, Visualization, Methodology, Writing - original draft, Project administration, Writing - review and editing; , Conceptualization, Resources, Data curation, Software, Formal analysis, Funding acquisition, Investigation, Writing - original draft, Writing - review and editing

### Author ORCIDs

Stéphanie Torrino https://orcid.org/0000-0002-8280-5907
Maeva Dufies http://orcid.org/0000-0003-1732-0388
Sonia Dagnino https://orcid.org/0000-0001-6846-7190
Delphine Debayle http://orcid.org/0000-0003-2807-9198
Frederic Bost https://orcid.org/0000-0003-4509-4701
Jerome Gilleron https://orcid.org/0000-0002-6389-8685

### Decision letter and Author response

Decision letter https://doi.org/10.7554/eLife.68348.sa1
Author response https://doi.org/10.7554/eLife.68348.sa2

## Additional files

### Supplementary files

• Supplementary file 1. Proteins identified by LC-MS/MS following UBTD1 immunoprecipitation in DU145 cell line.

• Supplementary file 2. Subcellular localization score of UBTD1 interactants in DU145 cell line.

• Transparent reporting form

### Data availability

All data generated or analysed during this study are included in the manuscript and supporting files. Source data files have been provided.

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
