## [Decision Letter]

**Acceptance summary:**

This study demonstrates that UBTD1 is a novel factor that regulates EGFR receptor function. Two mechanisms mediate the effects of UBTD1: (a) control of endolysosomal positioning and consequently EGFR degradation; and (b) control of ganglioside-mediated inhibition of EGFR tyrosine kinase activity. These findings advance our understanding of EGFR receptor function.

**Decision letter after peer review:**

[Editors’ note: the authors submitted for reconsideration following the decision after peer review. What follows is the decision letter after the first round of review.]

Thank you for submitting your work entitled ‘UBTD1 regulates ceramide balance and endolysosomal positioning to coordinate EGFR signaling’ for consideration by *eLife*. Your article has been reviewed by 3 peer reviewers, one of whom is a member of our Board of Reviewing Editors, and the evaluation has been overseen by a Senior Editor. The reviewers have opted to remain anonymous.

Our decision has been reached after consultation between the reviewers. While the reviewers found your study of the role of UBTD1 in the regulation of EGFR trafficking and tyrosine phosphorylation to be interesting, several aspects of your study require clarification and new experimentation to fully establish the proposed conclusions. The required changes are quite extensive and will take some time to address. We are therefore returning the manuscript to you at this time. Nevertheless, we would welcome the submission of a new manuscript that addresses the points raised during the review. We note that the reviewers are convinced that UBTD1 plays an important role in EGFR biology, but feel that the data supporting the proposed mechanism need improvement.

*Reviewer #1:*

1. The analysis of UBTD1 and ASAH1 is largely based on the use of single SMARTPool siRNAs and Control siRNAs. It is important that key observations are replicated in studies using different siRNA (either a different SMARTPool or using more than one single siRNA).

2. The phosphorylation data presented in Figure 1 would be improved by the inclusion of additional immunoblot analysis of proteins related to EGF signaling—for example, AKT & ERK. I do note that STAT3 phosphorylation is shown in Figure 1—figure supplement 1B/C, but it is not stated which site is examined; moreover, the increased phosphorylation detected in the immunoblot appears not to have been detected in the dot blot (Figure 1A/D).

3. Figure 1B does show immunoblot of the EGFR and EGFR pY1068 (note the dot blot shows a different site, EGFR Y-1086). Both EGFR and pEGFR are increased. These data should be quantitated (as was done for STAT3 in Figure 1—figure supplement 1C).

4. The EGF dependence of the changes in EGFR tyrosine phosphorylation. The conclusion presented is that it is EGF dependent. However, the dot blot (Figure 1 A) and the proliferation assay (Figure 1E) shows increases in the absence of EGF (therefore EGF-independent). In contrast to what is stated in the text of the manuscript, the EGFR inhibitor study (Figure 1G) provides no insight into the EGF-dependence of the observation.

5. The conclusion that UBTD1 regulates EGFR degradation is supported by the data presented. However, the mechanism is unclear. Data showing that SQSTM1/RNF26 and ubiquitination are altered by UBDT1, but is unclear whether this is mechanistically relevant to the observed changes in EGFR degradation.

*Reviewer #2:*

The paper by Torrino et al., describes the effects of UBTD1, which interacts with both E2 and E3 ubiquitin ligases, on EGFR phosphorylation, expression and trafficking. Overall, the effects shown appear to be robust and convincing. There are potential concerns about interpretation of specific data, and a general concern of question if the effects observed here are truly specific to EGFR. Detailed comments follow below.

1. Given that UBTD1 interacts with both E2 and E3 enzymes, and given the role reported here of UBTD1 in regulating membrane lipid composition, it seems likely that UBTD1 could be more broadly affecting the regulation of other receptor tyrosine kinases. Yet, the R&D phosphokinase array kit that was used by the investigators had a single receptor analyte within it. The authors should consider probing more broadly for effects on other receptors as such effects could also underlie differences in cellular proliferation rates such as those reported in Figure 1.

2. The PLA experiments shown in Figure 3B need appropriate controls, for example, with single antibodies.

3. The authors interpret their PLA results in Figure 3 as providing evidence of EGFR ubiquitination. PLA seems like an indirect way of showing that. Why have the authors not instead done an IP western blot, with pulldown for EGFR and blotting for ubiquitin?

4. In Figure 3D, the authors should show the baseline (t = 0 min) staining of the cells. It would be interesting to know, for example, if the elevated baseline phosphorylation is causing clustering and potential endocytosis even without EGF.

5. The authors refer to Figure 3D as showing evidence of internalization, but of course, on its own, clustering is not good evidence of endocytosis. The colocalization images shown in panels E and F are better for that purpose, and the authors should consider refining the text surround Figure 3D.

6. Given that the authors are looking in some cases at fairly long time points in their measurements, can they rule out other potential trafficking effects such as alterations in receptor recycling?

*Reviewer #3:*

This study provides evidence for a dual role of UBTD1 is regulating EGFR signalling, firstly in regulating autophosphorylation of the receptor via modulating ceramide levels and secondly via lysosomal degradation of the receptor.

UBTD1 is largely uncharacterised but has been postulated to scaffold ubiquitination complexes and have a tumour suppressor role. The current study supports this and identifies EGFR signalling as a novel target of UBTD1 complexes. Interestingly, it appears that distinct UBTD1 complexes act on different aspects of EGFR signalling. The role of UBTD1 in lysosome localisation is well characterised but the role in EGFR autophosphorylation less so.

1. UBTD1 knock-down does not increase cell proliferation in the presence of EGF (Figure 1E). However, EGFR (Tyr1086) phosphorylation is significantly increased after UBTD1 knock-down in the presence of EGF compared to cells treated with EGF and siRNA control (Figure 1D). Therefore, the increased phosphorylation of markers of EGFR activity as measured in the array in Figure 1D don't correlate with an increase in cell proliferation. Is there an explanation for this? It would maybe be clearer if immunoblots of EGFR, AKT, and STAT3 phosphorylation were included to support the array data (although STAT3 phospho-Tyr705 is not increased by EGF upon UBTD1 knock-down based on the array).

2. I don't follow the conclusion on pg5 based on Figure 1F that the pro-proliferative effect of UBTD1 knock-down is EGF-dependent. Gefitinib is an ATP-competitive inhibitor of EGFR so will inhibit its activity and downstream signalling irrespective of the mechanism by which EGFR is activated and whether this is EGF-dependent or EGF-independent. Also, the stated conclusion on pg14 is that UBTD1 depletion exacerbates EGFR signaling and induces EGF-dependent cellular proliferation. Based on Figure 1E shouldn't this be EGF-independent cellular proliferation?

3. A number of studies have shown that GM3 can suppress EGFR autophosphorylation and cell proliferation. From Figure 2E it seems that the addition of GM3 only supresses phospho-EGFR once UBTD1 is knocked-down (there seems to be a small increase in EGFR phosphorylation when GM3 is added to the siControl cells—why is this?).

(i) Does addition of GM3 suppress the increased proliferation induced by UBTD1 knock-down in the absence of EGF? As GM3 addition does not rescue EGFR protein levels, this might provide a guide to the relative importance of the two UBTD1 mechanisms in contributing to the regulation of cell proliferation.

(ii) There are also two Hsp90 immunoblots in this figure, one of which shows that Hsp90 levels decrease upon addition of GM3 after UBTD1 knock-down?

4. The authors propose that the decrease in GM3 levels upon UBTD1 knock-down is caused by an increase in the expression of the ceramidase ASAH1.

(i) It would be helpful to show that GM3 levels are increased upon siASAH1 in the DU145 cells.

(ii) It is unclear why p-EGFR levels increase upon knock-down of ASAH1 when UBTD1 is not knocked-down (Figure 2H). This could be explained by the siASAH1 also decreasing UBTD1 levels as indicated in the UBTD1 immunoblot, but it is unclear why ASAH1 knock-down will affect UBTD1 levels.

5. The evidence that UBTD1 knock-down increases p62 levels is weak. Supplementary Figure 4C suggests a slight increase but this is not really evident in Figure 4J or Supplementary Figure 4D. Do EGFR levels change upon RNF26 knock-down?

---

## [Author Response]

[Editors’ note: the authors resubmitted a revised version of the paper for consideration. What follows is the authors’ response to the first round of review.]

Reviewer #1:1. The analysis of UBTD1 and ASAH1 is largely based on the use of single SMARTPool siRNAs and Control siRNAs. It is important that key observations are replicated in studies using different siRNA (either a different SMARTPool or using more than one single siRNA).

Following the reviewer’s suggestion, we confirmed the effects of the UBTD1 ‘siRNA pool’ by using, in most key experiments (in parallel), different siRNAs directed against UBTD1 (siRNA single1 or single2) or ASAH1 (siRNA single1 or single2). New figures 3B, 3D, 6B, 6C, 6D, 6F, and Figure supplement 1B, 1E, 3A show that ‘single siRNA’ displays the same effect than the ‘siRNA pool’ on UBTD1 depletion associated effects.

2. The phosphorylation data presented in Figure 1 would be improved by the inclusion of additional immunoblot analysis of proteins related to EGF signaling – for example, AKT & ERK. I do note that STAT3 phosphorylation is shown in Fig1supp1B/C, but it is not stated which site is examined; moreover, the increased phosphorylation detected in the immunoblot appears not to have been detected in the dot blot (Figure 1A/D).

– Following the suggestion of the reviewer, we added, in complement to Figure 1 (Figure supplement 1D,E), western-blots that allow the reader to estimate the activation of the main signaling pathways downstream of the EGFR (Akt & ERK). We modified the text accordingly ‘Likewise, the phosphorylation of ERK (T202/Y204) and Akt (S473/T308) are increased in UBTD1-depleted cells’ (page 5).

– In the revised manuscript, we corrected all the western-blot figures by systematically mentioning the phosphorylation sites of the different proteins, including STAT3 (Figure 3D and Figure supplement 1D,E, 2E).

– We agree with the reviewer’s comment. Indeed, we observed a slight increase in STAT3 phosphorylation in the dot blot/Heatmap (< 1fold increase relative to the control) (Figure 1A). ‘Light blue’ (color code indicated on the scalebar) for p-STAT3 indicates an increase below 1-fold relative to the control. Consistent with this observation, there is an increase in p-STAT3 close to 50% (0.5-fold increase relative to the control) in the western blot (Figure supplement 1D) which we confirmed using single SiRNAs against UBTD1 (Figure supplement 1E).

3. Figure 1B does show immunoblot of the EGFR and EGFR pY1068 (note the dot blot shows a different site, EGFR Y-1086).

As recommended by the reviewer, in the revised manuscript, we now performed both p-Y1086 and pY1068 phosphorylation for all western-blot analysis (Figure 1B, 2E, 3D and Figure supplement 1B, 2E). Once activated by ligand binding and receptor dimerisation, the two phosphorylation sites (Y1068 and Y1086) activate the same signaling pathways (STAT3, ERK, and Akt by GRB2-gab1) and endocytosis process (mediated by CBL)^1^. As expected, after UBTD1 depletion, phosphorylation at both sites (Y1068 and Y1086) are increased using either a pool of siRNAs (siRNA Smartpool) or single siRNAs targeted against UBTD1.

^1^ Yao Huang and Yongchang Chang (November 25th 2011). Epidermal Growth Factor Receptor (EGFR) Phosphorylation, Signaling and Trafficking in Prostate Cancer, Prostate Cancer—From Bench to Bedside, Philippe E. Spiess, IntechOpen, DOI: 10.5772/27021.

Both EGFR and pEGFR are increased. These data should be quantitated (as was done for STAT3 in Fig1supp1C).

As recommended by the reviewer, we have now quantified the ratio of pY1068-EGFR/total EGFR and pY1086-EGFR/total EGFR (Figure 1B and Figure supplement 1B).

4. The EGF dependence of the changes in EGFR tyrosine phosphorylation. The conclusion presented is that it is EGF dependent. However, the dot blot (Figure 1 A) and the proliferation assay (Figure 1E) shows increases in the absence of EGF (therefore EGF-independent). In contrast to what is stated in the text of the manuscript, the EGFR inhibitor study (Figure 1G) provides no insight into the EGF-dependence of the observation.

We definitely agree with the reviewer's comment and apologise for the mistake. In the revised manuscript we have modified the text accordingly: ‘indicating that this effect is EGFR-dependent but EGF-independent’ (page 5).

5. The conclusion that UBTD1 regulates EGFR degradation is supported by the data presented. However, the mechanism is unclear. Data showing that SQSTM1/RNF26 and ubiquitination are altered by UBDT1, but is unclear whether this is mechanistically relevant to the observed changes in EGFR degradation.

This point is particularly relevant and is an essential part of our study. Thus, to reinforce our previous data, we performed a new experiment. As shown in Figure 6A-D, UBTD1 depletion inhibits p62 degradation (C), decreases p62 ubiquitination (A), and decreases the interactions between p62/RNF26 (D). To show that this process is causally responsible for the increase of EGFR level, we have invalidated RNF26 by siRNA. As shown in Figure 6E, we observe that RNF26 depletion increases the amount of p62 and EGFR. Interestingly, in UBTD1-depleted cells, RNF26 depletion does not have an additive effect on EGFR and p62 level, suggesting that UBTD1 and RNF26 act on the same molecular mechanism. From this new data, we added an extra paragraph in the text (page 10).

Reviewer #2:The paper by Torrino et al., describes the effects of UBTD1, which interacts with both E2 and E3 ubiquitin ligases, on EGFR phosphorylation, expression and trafficking. Overall, the effects shown appear to be robust and convincing.

We are very grateful to the reviewer for this positive comment.

There are potential concerns about interpretation of specific data, and a general concern of question if the effects observed here are truly specific to EGFR. Detailed comments follow below.1. Given that UBTD1 interacts with both E2 and E3 enzymes, and given the role reported here of UBTD1 in regulating membrane lipid composition, it seems likely that UBTD1 could be more broadly affecting the regulation of other receptor tyrosine kinases. Yet, the R&D phosphokinase array kit that was used by the investigators had a single receptor analyte within it.

The reviewer raises a very interesting point that we have been working on to clarify. To answer this legitimate question, we added a new supplemental figure (Figure supplement 4) and discussed the data in the text (page 9).

Briefly, we performed pulse-chase experiments with a fluorescently-labelled-HGF (ligand of c-Met receptor) to monitor the temporal kinetics of its degradation in the lysosome. As for the EGF, in UBTD1-depleted cells, we observe a delay in the extinction of the intracellular fluorescent-HGF signal (Figure supplement 4A). Next, we tested the effect of UBTD1 knock-down on the ubiquitination of c-Met. As with EGFR, we found that UBTD1 depletion does not alter c-Met ubiquitination (Figure supplement 4B-C). Lastly, we wanted to know if, as for the EGFR, the depletion of UBTD1 affects the total amount of c-Met (Figure supplement 4D). Indeed, in UBTD1-depleted cells, we observe an increase in the level of c-Met and TGF-β R without any change in their mRNA expression (Figure supplement 4E). Nevertheless, this mechanism seems to be restricted to tyrosine kinase receptors since we do not observe any change in the amount of IL2 or Transferrin receptors (Figure supplement 4D).

We also modified the discussion accordingly: ‘Importantly, this process is presumably not restricted to EGFR since we observe, as anticipated by Jongsma et al. [33], similar effects of UBTD1 depletion on other signaling receptors that use the same intracellular routing as EGFR such as c-Met or TGF-β R [40, 41]’ (page 9).

The authors should consider probing more broadly for effects on other receptors as such effects could also underlie differences in cellular proliferation rates such as those reported in Figure 1.

Although, as proposed by the reviewer, it is clear that UBTD1 depletion does not only affect EGFR degradation but also others tyrosine kinase receptors, the effect of UBTD1 depletion on cell proliferation is strictly related to EGFR. Indeed, the use of Gefinitib totally abrogates the proproliferative effects induced by UBTD1 depletion (Figure 1F). Moreover, as we have shown previously (Torrino S. et al., 2019)^2^ and described earlier by others (Miura H. et al., 2001)^3^, cMet/HGF does not affect proliferation in DU145 cells (prostate cancer cell line) and TGFβ-R is known to inhibit proliferation (Wilding G. et al., 1989)^4^.

^2^Torrino S, Roustan FR, Kaminski L, Bertero T, Pisano S, Ambrosetti D, Dufies M, Uhler JP, Lemichez E, Mettouchi A, Gesson M, Laurent K, Gaggioli C, Michiels JF, Lamaze C, Bost F, Clavel S. UBTD1 is a mechano-regulator controlling cancer aggressiveness. EMBO Rep. 2019 20(4):e46570.

^3^H Miura , K Nishimura, A Tsujimura, K Matsumiya, K Matsumoto, T Nakamura, A Okuyama. Effects of hepatocyte growth factor on E-cadherin-mediated cell-cell adhesion in DU145 prostate cancer cells. Urology, 2001 58(6):1064–9.

^4^G Wilding, G Zugmeier, C Knabbe, K Flanders, E Gelmann. Differential effects of transforming growth factor β on human prostate cancer cells in vitro. Mol Cell Endocrinol. 1989 ;62(1):79–87.

2. The PLA experiments shown in Figure 3B need appropriate controls, for example, with single antibodies.

We added the appropriate controls (single antibody) in all PLA experiments including Figure 4B, 6D and Figure supplement 4B, 6C.

3. The authors interpret their PLA results in Figure 3 as providing evidence of EGFR ubiquitination. PLA seems like an indirect way of showing that. Why have the authors not instead done an IP western blot, with pulldown for EGFR and blotting for ubiquitin?

To confirm that UBTD1 depletion does not modify EGFR or c-Met ubiquitination, we have followed the reviewer’s suggestion and added new IP experiments for Ub-EGFR (Figure supplement 3B) and Ubc-MET (Figure supplement 4C).

4. In Figure 3D, the authors should show the baseline (t = 0 min) staining of the cells. It would be interesting to know, for example, if the elevated baseline phosphorylation is causing clustering and potential endocytosis even without EGF.

We now added the baseline (t = 0 min) staining of the cells for the previous Figure 3D (New Figure 4D). Because, the time=0 of the pulse-chase experiment corresponds to a condition in presence of EGFalexa647 bound at 4°C, under this condition, EGF is already present in the media.

5. The authors refer to Figure 3D as showing evidence of internalisation, but of course, on its own, clustering is not good evidence of endocytosis. The colocalisation images shown in panels E and F are better for that purpose, and the authors should consider refining the text surround Figure 3D.

We agree with the reviewer and therefore we changed the text accordingly:

‘The amount of internalised EGF remained constant during the first 30 min in both control and UBTD1 knock-down cells, suggesting that the internalisation remain similar’ and ‘The arrival and departure of EGF in EEA1-positive compartments was similar in UBTD1-depleted and control cells (Figure 4E), confirming that UBTD1 depletion does not alter endocytosis and the first steps of EGFR intracellular trafficking’ (page 8).

6. Given that the authors are looking in some cases at fairly long time points in their measurements, can they rule out other potential trafficking effects such as alterations in receptor recycling?

The reviewer is correct that in some cases we analysed long time points, but this was restricted to the pulse-chase experiments where we followed either the decay of EGF-alexa647 or the colocalisation of EGF-alexa647 with EEA1 or LAMP1 loaded as a single bulk of labelled EGF at time=0. Therefore contribution of newly internalised EGFR/ ‘unlabelled EGF’ after the time=0 due to a potential change in EGFR recycling may be excluded. Moreover, a dramatic change in EGFR recycling will undoubtly change the quantity of EGF receptor at the cell surface, which we did not observe (Author response image 1) confirming pulse-chase (Figure 4D) and Flow-cytometry experiments (Figure 4C).

**Author response image 1. respfig1:** Staining of surface EGFR using an antibody against the extracellular region of EGFR bound at 4°C in unpermeabilised. Control and UBTD1-depleted cells.

However, the reviewer is correct that a change in recycling of EGFR bound to its ligand back to the plasma membrane may also change the load of EGF-alexa647, similarly to what is well known and often tested for the transferrin receptor. Nevertheless, EGFR was never found to highly recycle together with its ligand (on the opposite to the transferrin receptor), however, to exclude such a possibility raised by the reviewer we analysed the quantity of EGF-alexa647 co-localised with TGN46 in control and UBTD1-depleted cells. We observed a very low colocalisation between these two markers, and the rate of colocalisation was similar in between both conditions excluding a dramatic contribution of recycling (Author response image 2).

**Author response image 2. respfig2:** Cells were treated with EGF-647 (100 ng/ml) and stained with a Trans-Golgi network marker (TGN46) to evaluate recycling of EGF-647 in control and UBTD1-depleted cells.

Reviewer #3:This study provides evidence for a dual role of UBTD1 is regulating EGFR signalling, firstly in regulating autophosphorylation of the receptor via modulating ceramide levels and secondly via lysosomal degradation of the receptor.UBTD1 is largely uncharacterised but has been postulated to scaffold ubiquitination complexes and have a tumour suppressor role. The current study supports this and identifies EGFR signalling as a novel target of UBTD1 complexes. Interestingly, it appears that distinct UBTD1 complexes act on different aspects of EGFR signalling.

We are very grateful to the reviewer for this highly positive judgement.

The role of UBTD1 in lysosome localisation is well characterised but the role in EGFR autophosphorylation less so.1. UBTD1 knock-down does not increase cell proliferation in the presence of EGF (Figure 1E). However, EGFR (Tyr1086) phosphorylation is significantly increased after UBTD1 knock-down in the presence of EGF compared to cells treated with EGF and siRNA control (Figure 1D). Therefore, the increased phosphorylation of markers of EGFR activity as measured in the array in Figure 1D don't correlate with an increase in cell proliferation. Is there an explanation for this?

As noted by the reviewer, UBTD1 depletion strongly increases cell proliferation (40%) which probably reaches a maximum threshold since even high concentrations of EGF do not increase it further (Figure 1E). The effect of UBTD1 depletion on cell proliferation is strictly related to EGFR since the use of Gefinitib totally abrogates this effect (Figure 1F).

The two experiments to which the reviewer refers are hardly comparable (Figures 1E and 1D). Indeed, in the ‘proliferation experiment’ (Figure 1E), the cells were cultivated with EGF for 48h to ‘see the effect’ of EGF on cell proliferation. On the contrary, in the ‘signaling experiment’ (Figure 1D), the cells were acutely treated with EGF (10 min, 50 ng/ml), to get a maximal signal response and to avoid inhibitory feedback and receptor degradation that take place over time (h).

It would maybe be clearer if immunoblots of EGFR, AKT and STAT3 phosphorylation were included to support the array data (although STAT3 phospho-Tyr705 is not increased by EGF upon UBTD1 knock-down based on the array).

We agree with reviewer 1 (question 2) and 3 and provided additional immunoblot analysis as recommended. Please refer to response elements in the reply to reviewer 1 (question 2) for a complete list of modified figures.

– Following the suggestion of the reviewer, we added, in complement to Figure 1 (Figure supplement 1D-E), western-blots that allow the reader to estimate the activation of the main signaling pathways downstream of the EGFR (Akt and ERK). We modified the text accordingly ‘Likewise, the phosphorylation of ERK (T202/Y204) and Akt (S473/T308) are increased in UBTD1-depleted cells’ (page 5).

– We agree with the reviewer’s comment. Indeed, we observed a slight increase in STAT3 phosphorylation in the dot blot/Heatmap (< 1fold increase relative to the control) (Figure 1A). ‘Light blue’ (color code indicated on the scalebar) for p-STAT3 indicates an increase below 1-fold relative to the control. consistent with this observation, there is an increase in p-STAT3 close to 50% (0.5-fold increase relative to the control) in the western blot (Figure supplement 1D) which we confirmed using single SiRNAs against UBTD1 (Figure supplement 1E).

2. I don't follow the conclusion on pg5 based on Figure 1F that the pro-proliferative effect of UBTD1 knock-down is EGF-dependent. Gefitinib is an ATP-competitive inhibitor of EGFR so will inhibit its activity and downstream signalling irrespective of the mechanism by which EGFR is activated and whether this is EGF-dependent or EGF-independent. Also, the stated conclusion on pg14 is that UBTD1 depletion exacerbates EGFR signaling and induces EGF-dependent cellular proliferation. Based on Figure 1E shouldn't this be EGF-independent cellular proliferation?

We definitely agree with the reviewer's comment and apologise for the mistake. In the revised manuscript we have modified the text accordingly: ‘indicating that this effect is EGFR-dependent but EGF-independent’ (page 5) and ‘UBTD1 depletion exacerbates EGFR signaling and induces an EGFR-dependent cellular proliferation’ (page 15).

3. A number of studies have shown that GM3 can suppress EGFR autophosphorylation and cell proliferation. From Figure 2E it seems that the addition of GM3 only supresses phospho-EGFR once UBTD1 is knocked-down (there seems to be a small increase in EGFR phosphorylation when GM3 is added to the siControl cells—why is this?).

To verify our previous result, we performed western-blot and quantification for both Y1068 and Y1086 phosphorylation sites of EGFR. As depicted and quantified in the new Figure 2E, GM3 does not increase EGFR phosphorylation (siControl) while GM3 treatment decrease EGFR phosphorylation (Y1068/Y1086) in UBTD1-depleted cells.

(i) Does addition of GM3 suppress the increased proliferation induced by UBTD1 knock-down in the absence of EGF? As GM3 addition does not rescue EGFR protein levels, this might provide a guide to the relative importance of the two UBTD1 mechanisms in contributing to the regulation of cell proliferation.

To answer this question, we carried out a new proliferation experiment in the presence of GM3 (Figure 2F). As anticipated by the reviewer, the experiment gives a particularly clear result. Indeed, GM3 treatment inhibits the increased proliferation induced by UBTD1 knock-down. As mentioned by the reviewer, it is an important clue to explain the mechanism by which UBTD1 controls proliferation. We modify the text accordingly: ‘In UBTD1-depleted cells, addition of GM3 restores, in a dose dependent manner, the level of EGFR phosphorylation observed in control cells and completely abolishes cell proliferation induced by UBTD1 knock-down (Figure 2E, F), suggesting that the increase in EGFR phosphorylation and cell proliferation was due to a drop in GM3’ (page 6).

(ii) There are also two Hsp90 immunoblots in this figure, one of which shows that Hsp90 levels decrease upon addition of GM3 after UBTD1 knock-down?

We re-performed the western-blot and quantified it in the new Figure 2E, Hsp90 levels did not decrease upon addition of GM3 after UBTD1 knock-down

4. The authors propose that the decrease in GM3 levels upon UBTD1 knock-down is caused by an increase in the expression of the ceramidase ASAH1.(i) It would be helpful to show that GM3 levels are increased upon siASAH1 in the DU145 cells.

At the request of the reviewer, we performed a GM3 content measurement (Elisa assay) in ASAH1 depleted cells (siRNA). As expected ASAH1 depletion increases GM3 content (Author response image 3).

**Author response image 3. respfig3:** GM3 levels are increased upon siASAH1 in the DU145 cells.

(ii) It is unclear why p-EGFR levels increase upon knock-down of ASAH1 when UBTD1 is not knocked-down (Figure 2H). This could be explained by the siASAH1 also decreasing UBTD1 levels as indicated in the UBTD1 immunoblot, but it is unclear why ASAH1 knock-down will affect UBTD1 levels.

We re-perform the western-blot (siRNA pools) and quantified it (Figure supplement 2E). In addition, we performed a new set of experiment with siRNA against UBTD1 (siRNA single1, siRNA single2), control (siRNA single1), ASAH1 (siRNA single1, siRNA single2) (Figure 3D). As shown in Figure 3D and Figure supplement 2E, ASAH1 knock-down (siRNA pool or singles) does not modify neither EGFR phosphorylation (Y1068/Y1086) nor UBTD1 levels.

5. The evidence that UBTD1 knock-down increases p62 levels is weak. Supplementary Figure 4C suggests a slight increase but this is not really evident in Figure 4J or Supplementary Figure 4D.

As suggested by the reviewer, to strengthen our initial assumption on the connection between UBTD1 and p62:

– We completed our previous results using two single siRNAs against UBTD1 (Figure 6B). As seen in Figure 6B, UBTD1 knock-down increases p62 levels.

– To complement our previous ‘p62 degradation assay’ (Figure supplement 6B), we performed new experiments with two single siRNA against UBTD1 (Figure 6C). In the UBTD1-depleted cells (single siRNA), we observed an increase in p62 protein lifetime compared to control cells (Figure 6C).

– We also confirmed our previous PLA experiments between p62 and RNF26 (Figure supplement 6C) by using single siRNA against UBTD1 (Figure 6D).

Collectively, our new set of experiments confirms our original proposal that, UBTD1 depletion inhibits p62 degradation (Figure 6C—Figure supplement 6B), decreases p62 ubiquitination (Figure 6A), decreases the interactions between p62/RNF26 (Figure 6D—Figure supplement 6C) and increase p62 content (Figure 6B—Figure supplement 6A).

Do EGFR levels change upon RNF26 knock-down?

To answer this question, we performed a new experiment (Figure 6E). As clearly shown in Figure 6E, the answer is ‘yes’. We observe that RNF26 depletion increases the amount of p62 and EGFR. Interestingly, in UBTD1-depleted cells, RNF26 depletion does not have an additive effect on EGFR and p62 level, suggesting that UBTD1 and RNF26 act on the same molecular mechanism. From this new data, we added an extra paragraph in the text (page 10).